# AlignDiff: Aligning Diverse Human Preferences via Behavior-Customisable Diffusion Model

**Zibin Dong**[*1], **Yifu Yuan**[*1], **Jianye Hao**[†1], **Fei Ni**[1], **Yao Mu**[3], **Yan Zheng**[1],
**Yujing Hu**[2], **Tangjie Lv**[2], **Changjie Fan**[2], **Zhipeng Hu**[2]
[1]College of Intelligence and Computing, Tianjin University,
[2]Fuxi AI Lab, Netease, Inc., Hangzhou, China, [3]The University of Hong Kong

## ABSTRACT

Aligning agent behaviors with diverse human preferences remains a challenging problem in reinforcement learning (RL), owing to the inherent *abstractness* and *mutability* of human preferences. To address these issues, we propose **Align-Diff**, a novel framework that leverages RLHF to quantify human preferences, covering *abstractness*, and utilizes them to guide diffusion planning for zero-shot behavior customizing, covering *mutability*. AlignDiff can accurately match user-customized behaviors and efficiently switch from one to another. To build the framework, we first establish the multi-perspective human feedback datasets, which contain comparisons for the attributes of diverse behaviors, and then train an attribute strength model to predict quantified relative strengths. After relabeling behavioral datasets with relative strengths, we proceed to train an attribute-conditioned diffusion model, which serves as a planner with the attribute strength model as a director for preference aligning at the inference phase. We evaluate AlignDiff on various locomotion tasks and demonstrate its superior performance on preference matching, switching, and covering compared to other baselines. Its capability of completing unseen downstream tasks under human instructions also showcases the promising potential for human-AI collaboration. More visualization videos are released on https://aligndiff.github.io/.

## 1 INTRODUCTION

One of the major challenges in building versatile RL agents is aligning their behaviors with human preferences (Han et al., 2021; Guan et al., 2023). This is primarily due to the inherent *abstractness* and *mutability* of human preferences. The *abstractness* makes it difficult to directly quantify preferences through a hand-designed reward function (Hadfield-Menell et al., 2017; Guan et al., 2022), while the *mutability* makes it challenging to design a one-size-fits-all solution since preferences vary among individuals and change over time (Pearce et al., 2023). Addressing these two challenges can greatly enhance the applicability and acceptance of RL agents in real life.

The *abstractness* of human preferences makes manual-designed reward functions not available (Vecerik et al., 2018; Hadfield-Menell et al., 2017). Other sources of information such as image goals or video demonstrations are being incorporated to help agents understand preferred behaviors (Andrychowicz et al., 2017; Pathak et al., 2018; Ma et al., 2022). However, expressing preferences through images or videos is inconvenient for users. Expressing through natural language is a more user-friendly option. Many studies have investigated how to map natural languages to reward signals for language-guided learning (Goyal et al., 2019; Arumugam et al., 2017; Misra et al., 2017). However, due to language ambiguity, grounding agent behaviors becomes difficult, leading to limited effective language instructions. One promising approach is to deconstruct the agent's behavior into combinations of multiple Relative Behavioral Attributes (RBA) at varying levels of strength (Guan et al., 2023), allowing humans to express preferences in terms of relative attribute strengths.

---

[*]These authors contributed equally to this work.
[†]Corresponding author: Jianye Hao (jianye.hao@tju.edu.cn)

The primary limitation of this approach is that it merely refines attribute evaluation into a myopic, single-step state-action reward model, ignoring the impact on overall trajectory performance.

The *mutability* of human preferences emphasizes the need for zero-shot behavior customizing. Training agents using an attribute-conditioned reward model fails to overcome this issue (Hu et al., 2020; Guan et al., 2023), as it requires constant fine-tuning or even retraining whenever the user's intention changes, resulting in a poor user experience. To bridge this gap, we need a model to fit diverse behaviors and support retrieving based on the relative attribute strengths without repetitive training. Although standard choices like conditional behavior cloning can achieve similar goals, they are limited by poor expressiveness and fail to capture diverse human preferences (Ross et al., 2011; Razavi et al., 2019; Orsini et al., 2021). Therefore, we focus on powerful conditional generative models, specifically diffusion models, which have demonstrated excellent expressiveness in dealing with complex distributions and superior performance in decision-making tasks with complex dynamics (Janner et al., 2022; Ajay et al., 2023; Dong et al., 2024).

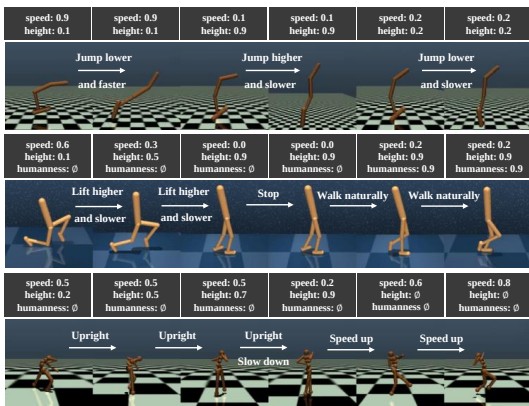

In this paper, we introduce AlignDiff, a novel framework that leverages RLHF to quantify human preferences, covering *abstractness*, and utilizes them to guide diffusion planning for zero-shot behavior customizing, covering *mutability*. We created multi-perspective human feedback datasets containing comparisons for the attributes on diverse behaviors, which were used to train a transformer-based attribute strength model. This model captures the relative strength of attributes on the trajectory level. We then used the attribute strength model to annotate behavioral datasets and trained a diffusion model for planning. Within AlignDiff, agents can accurately match user-customized behaviors and efficiently switch from one to another. Its name, AlignDiff, represents both **Align**ment **Diff**usion and **align**ing to bridge the **diff**erence between human preferences and agent behaviors. We summarize the main contributions of AlignDiff as follows:

Figure 1: AlignDiff achieves zero-shot human preferences aligning. Here, three robots continuously switch their behaviors based on human instructions, where $\phi$ represents no effect.

- We introduce AlignDiff: a novel framework that leverages RLHF technique to quantify human preferences and utilizes them to guide diffusion planning for zero-shot behavior customizing.
- We establish reusable multi-perspective human feedback datasets through crowdsourcing, which contains diverse human judgments on relative strengths of pre-defined attributes for various tasks. By making our dataset repositories publicly available, we aim to contribute to the wider adoption of human preference aligning.
- We design a set of metrics to evaluate an agent's preference matching, switching, and covering capability, and evaluate AlignDiff on various locomotion tasks. The results demonstrate its superior performance in all these aspects. Its capability of completing unseen downstream tasks under human instructions also showcases the promising potential for human-AI collaboration.

## 2 RELATED WORKS

### 2.1 REINFORCEMENT LEARNING FROM HUMAN FEEDBACK

Reinforcement Learning from Human Feedback (RLHF) is a powerful technique that can speed up AI training and enhance AI capabilities by leveraging human feedback. (Pilarski et al., 2011; Akrour et al., 2011; Wilson et al., 2012; Akrour et al., 2012; Wirth & Fürnkranz, 2013; Zhou et al., 2024; Liu et al., 2024; Yuan et al., 2024). There are various forms of human feedback (Bıyık et al., 2022; Cabi et al., 2019; Lee et al., 2021), in which collecting pairwise comparison over decision trajectories is a common approach. This feedback is used to learn reward models for RL training, which significantly improves training efficiency and performance, especially in environments with sparse or ill-defined rewards (Christiano et al., 2017; Lee et al., 2021; Park et al., 2022; Liang et al., 2023a; Zhou et al., 2020). However, their limitation lies in the focus on optimizing a single objective. In

recent years, RLHF has also been applied to fine-tune large language models (LLMs) by leveraging human preferences to improve truthfulness and reduce toxic outputs (Stiennon et al., 2020; Ouyang et al., 2022). These methods highlight its potential for human preference quantification. RBA leverages RLHF to distill human understanding of abstract preferences into an attribute-conditioned reward model, which achieves simple and effective quantification. However, this approach has limitations. It refines the evaluation of attributes into a single step and ignores the impact on the overall trajectory. Furthermore, it requires retraining whenever new preferences emerge.

## 2.2 DIFFUSION MODELS FOR DECISION MAKING

Diffusion models, a type of score matching-based generative model (Sohl-Dickstein et al., 2015; Ho et al., 2020), initially gained popularity in the field of image generation (Ramesh et al., 2021; Galatolo et al., 2021; Saharia et al., 2022; Rombach et al., 2022). Their strong conditional generation capabilities have led to success in various domains (Peng et al., 2023; Rasul et al., 2021; Reid et al., 2023; Liu et al., 2023), including decision making (Janner et al., 2022; Liang et al., 2023b; Chen et al., 2023; Wang et al., 2023; Pearce et al., 2023; Hegde et al., 2023; Dong et al., 2024). One common approach is employing diffusion models to generate decision trajectories conditioned on rewards or other auxiliary information, which are then used for planning (Ajay et al., 2023; Ni et al., 2023). However, these reward-conditioned generations only utilize a small portion of the learned distribution. We propose using human preferences to guide diffusion models to discover and combine diverse behaviors, ensuring the full utilization of the learned trajectory distribution.

## 3 PRELIMINARIES

**Problem setup:** We consider the scenario where human users may want to customize an agent's behavior at any given time. This problem can be framed as a reward-free Markov Decision Process (MDP) denoted as $\mathcal{M} = \langle S, A, P, \boldsymbol{\alpha} \rangle$. Here, $S$ represents the set of states, $A$ represents the set of actions, $P : S \times A \times S \to [0, 1]$ is the transition function, and $\boldsymbol{\alpha} = \{\alpha_1, \cdots, \alpha_k\}$ represents a set of $k$ predefined attributes used to characterize the agent's behaviors. Given a state-only trajectory $\tau^l = \{s_0, \cdots, s_{l-1}\}$, we assume the existence of an attribute strength function that maps the trajectory to a relative strength vector $\zeta^{\boldsymbol{\alpha}}(\tau^l) = \boldsymbol{v}^{\boldsymbol{\alpha}} = [v^{\alpha_1}, \cdots, v^{\alpha_k}] \in [0, 1]^k$. Each element of the vector indicates the relative strength of the corresponding attribute. A value of 0 for $v^{\alpha_i}$ implies the weakest manifestation of attribute $\alpha_i$, while a value of 1 represents the strongest manifestation. We formulate human preferences as a pair of vectors $(\boldsymbol{v}^{\boldsymbol{\alpha}}_{\text{targ}}, \boldsymbol{m}^{\boldsymbol{\alpha}})$, where $\boldsymbol{v}^{\boldsymbol{\alpha}}_{\text{targ}}$ represents the target relative strengths, and $\boldsymbol{m}^{\boldsymbol{\alpha}} \in \{0, 1\}^k$ is a binary mask indicating which attributes are of interest. The objective is to find a policy $a = \pi(s|\boldsymbol{v}^{\boldsymbol{\alpha}}_{\text{targ}}, \boldsymbol{m}^{\boldsymbol{\alpha}})$ that minimizes the L1 norm $||(\boldsymbol{v}^{\boldsymbol{\alpha}}_{\text{targ}} - \zeta^{\boldsymbol{\alpha}}(\mathbb{E}_\pi[\tau^l])) \circ \boldsymbol{m}^{\boldsymbol{\alpha}}||_1$, where $\circ$ denotes the Hadamard product. We learn human preferences from an unlabeled state-action dataset $\mathcal{D} = \{\tau\}$, which contains multiple behaviors.

**Preference-based Reinforcement Learning (PbRL)** (Christiano et al., 2017) is a framework that leverages human feedback to establish reward models for RL training. In PbRL, researchers typically collect pairwise feedback from annotators on decision trajectories $(\tau_1, \tau_2)$. Feedback labels are represented as $y \in \{(1, 0), (0, 1), (0.5, 0.5)\}$, where $(1, 0)$ indicates that $\tau_1$ performs better, $(0, 1)$ indicates that $\tau_2$ performs better, and $(0.5, 0.5)$ indicates comparable performance. The collected feedback dataset is denoted as $\mathcal{D}_p = \{\tau_1, \tau_2, y\}$ and can be used to train reward models.

**Denoising Diffusion Implicit Models (DDIM)** (Song et al., 2021) are a type of diffusion model that consists of a non-Markovian forward process and a Markovian reverse process for learning the data distribution $q(\boldsymbol{x})$. The forward process $q(\boldsymbol{x}_t|\boldsymbol{x}_{t-1}, \boldsymbol{x}_0)$ is non-Markovian but designed to ensure that $q(\boldsymbol{x}_t|\boldsymbol{x}_0) = \mathcal{N}(\sqrt{\xi_t}\boldsymbol{x}_0, (1 - \xi_t)\boldsymbol{I})$. The reverse process $p_\phi(\boldsymbol{x}_{t-1}|\boldsymbol{x}_t) := \mathcal{N}(\mu_\phi(\boldsymbol{x}_t, t), \Sigma_t)$ is trained to approximate the inverse transition kernels. Starting with Gaussian noise $\boldsymbol{x}_T \sim \mathcal{N}(\boldsymbol{0}, \boldsymbol{I})$, a sequence of latent variables $(\boldsymbol{x}_{T-1}, \cdots, \boldsymbol{x}_1)$ is generated iteratively through a series of reverse steps using the predicted noise. The noise predictor $\epsilon_\phi(\boldsymbol{x}_t, t)$, parameterized by a deep neural network, estimates the noise $\epsilon \sim \mathcal{N}(\boldsymbol{0}, \boldsymbol{I})$ added to the dataset sample $\boldsymbol{x}_0$ to produce the noisy sample $\boldsymbol{x}_t$. Using DDIM, we can choose a short subsequence $\kappa$ of length $S$ to perform the reverse process, significantly enhancing sampling speed.

**Classifier-free guidance (CFG)** (Ho & Salimans, 2021) is a guidance method for conditional generation, which requires both a conditioned noise predictor $\epsilon_\phi(\boldsymbol{x}_t, t, \boldsymbol{c})$ and an unconditioned one

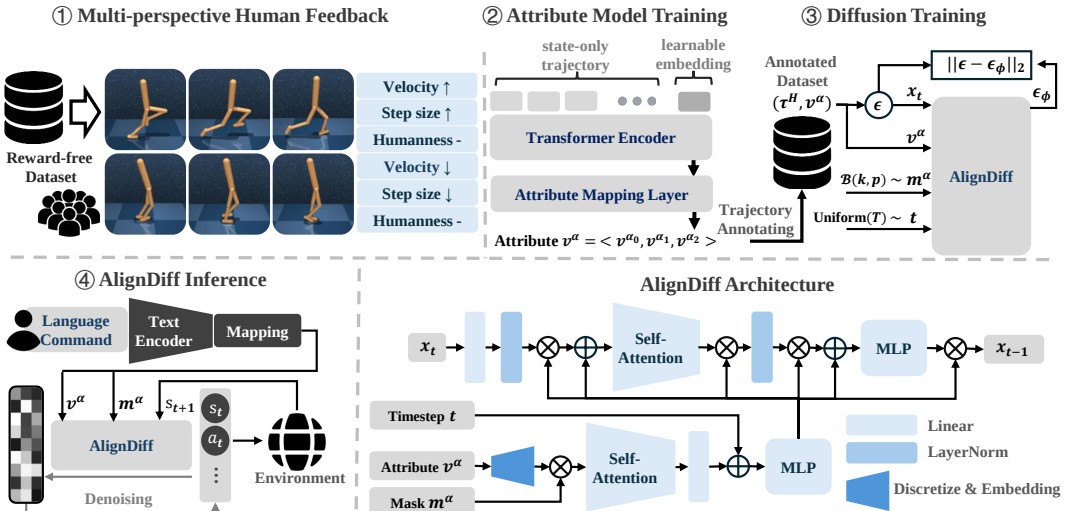

Figure 2: Overview of AlignDiff. We begin by collecting human feedback through crowdsourcing, which is then used to train an attribute strength model $\hat{\zeta}_\theta^\alpha$. Relabeled by it, an annotated dataset is then used to train an attribute-conditioned diffusion model $\epsilon_\phi$. With these two components, we can use AlignDiff to conduct preference alignment planning.

$\epsilon_\phi(\boldsymbol{x}_t, t)$, where $\boldsymbol{c}$ is the condition variable. By setting a guidance scale $w$ and giving a condition $\boldsymbol{c}$, we use $\tilde{\epsilon}_\phi(\boldsymbol{x}_t, t, \boldsymbol{c}) = (1+w)\epsilon_\phi(\boldsymbol{x}_t, t, \boldsymbol{c}) - w\epsilon_\phi(\boldsymbol{x}_t, t)$ to predict noise during the reverse process.

## 4 METHODOLOGY

We propose AlignDiff, a novel framework that leverages RLHF to quantify human preferences and utilizes them to guide diffusion planning for zero-shot behavior customizing. The process of generalization is depicted in Fig. 2, and the framework consists of four parts. Firstly, we collect multi-perspective human feedback through crowdsourcing. Secondly, we use this feedback to train an attribute strength model, which we then use to relabel the behavioral datasets. Thirdly, we train a diffusion model on the annotated datasets, which can understand and generate trajectories with various attributes. Lastly, we can use AlignDiff for inference, aligning agent behaviors with human preferences at any time. Details of each part are discussed in the following sections.

### 4.1 MULTI-PERSPECTIVE HUMAN FEEDBACK COLLECTION

We extract a small subset of pairs $\{(\tau_1, \tau_2)\}$ from the behavioral datasets $\mathcal{D}$ and ask human annotators to provide pairwise feedback. Instead of simply asking them to choose a better one between two videos, we request analyzing the relative performance of each attribute $\alpha_i$, resulting in a feedback dataset $\mathcal{D}_p = \{(\tau_1, \tau_2, y_{\text{attr}})\}$, where $y_{\text{attr}} = (y_1, \cdots, y_k)$. For example (see Fig. 2, part ①), to evaluate a bipedal robot with attributes (speed, stride, humanness), the annotators would be provided with two videos and asked to indicate which one has higher speed/bigger stride/better humanness, respectively. This process is crucial for creating a model that can quantify relative attribute strength, as many attributes, such as humanness, can only be evaluated through human intuition. Even for more specific and measurable attributes, such as speed, using human labels helps the model produce more distinct behaviors, as shown by our experiments.

### 4.2 ATTRIBUTE STRENGTH MODEL TRAINING

After collecting the feedback dataset, we can train the attribute strength model by optimizing a modified Bradley-Terry objective (Bradley & Terry, 1952). We define the probability that human annotators perceive $\tau_1$ to exhibit a stronger performance on $\alpha_i$ as follows:

$$P^{\alpha_i}[\tau_1 \succ \tau_2] = \frac{\exp \hat{\zeta}_{\theta,i}^{\boldsymbol{\alpha}}(\tau_1)}{\sum_{j \in \{1,2\}} \exp \hat{\zeta}_{\theta,i}^{\boldsymbol{\alpha}}(\tau_j)} \tag{1}$$

where $\hat{\zeta}_{\theta,i}^{\boldsymbol{\alpha}}(\tau)$ indicates the $i$-th element of $\hat{\zeta}_\theta^{\boldsymbol{\alpha}}(\tau)$. To approximate the attribute strength function, we optimize the following modified Bradley-Terry objective:

$$\mathcal{L}(\hat{\zeta}_\theta^{\boldsymbol{\alpha}}) = - \sum_{(\tau_1,\tau_2,y_{\text{attr}})\in D_p} \sum_{i\in\{1,\cdots,k\}} y_i(1)\log P^{\alpha_i}[\tau_1 \succ \tau_2] + y_i(2)\log P^{\alpha_i}[\tau_2 \succ \tau_1] \qquad (2)$$

It's worth noting that there are significant differences between learning the attribute strength model and the reward model in RLHF. For instance, one-step state-action pairs cannot capture the attribute strengths, so $\hat{\zeta}_\theta^{\boldsymbol{\alpha}}$ must be designed as a mapping concerning trajectories. Additionally, we aim for $\hat{\zeta}_\theta^{\boldsymbol{\alpha}}$ to accommodate variable-length trajectory inputs. Therefore, we use a transformer encoder as the structure (see Fig. 2, part ②), which includes an extra learnable embedding concatenated with the input. The output corresponding to this embedding is then passed through a linear layer to map it to the relative strength vector $\boldsymbol{v}^{\boldsymbol{\alpha}}$. After training, we can partition the dataset $\mathcal{D}$ into fixed-length trajectories of length $H$. These trajectories are then annotated using $\hat{\zeta}_\theta^{\boldsymbol{\alpha}}$, resulting in a dataset $\mathcal{D}_G = \{(\tau^H, \boldsymbol{v}^{\boldsymbol{\alpha}})\}$ for diffusion training.

## 4.3 DIFFUSION TRAINING

We use relative strength values $\boldsymbol{v}^{\boldsymbol{\alpha}}$ and attribute masks $\boldsymbol{m}^{\boldsymbol{\alpha}}$ as conditioning inputs $\boldsymbol{c}$ for the diffusion model, resulting in a conditioned noise predictor $\epsilon_\phi(\boldsymbol{x}_t, \boldsymbol{v}^{\boldsymbol{\alpha}}, \boldsymbol{m}^{\boldsymbol{\alpha}})$ and an unconditioned one $\epsilon_\phi(\boldsymbol{x}_t)$. According to our definition of $\boldsymbol{m}^{\boldsymbol{\alpha}}$, $\epsilon_\phi(\boldsymbol{x}_t)$ is equivalent to a conditioned noise predictor with a mask where all values are 0. Therefore, we only need one network to represent both types of noise predictors simultaneously. However, the network structure must meet two requirements: 1) $\boldsymbol{m}^{\boldsymbol{\alpha}}$ should eliminate the influence of nonrequested attributes on the model while preserving the effect of the interested attributes, and 2) $\boldsymbol{v}^{\boldsymbol{\alpha}}$ cannot be simply multiplied with $\boldsymbol{m}^{\boldsymbol{\alpha}}$ and fed into the network, as a value of 0 in $\boldsymbol{v}^{\boldsymbol{\alpha}}$ still carries specific meanings. To meet these requirements, we design an attribute-oriented encoder. First, we discretize each dimension of the relative strength vector into $V$ selectable tokens as follows:

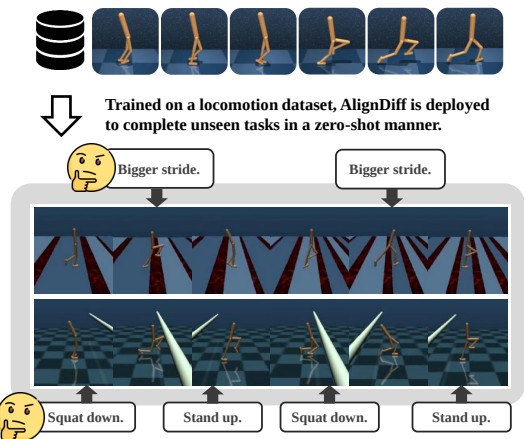

Figure 3: AlignDiff, trained solely on a locomotion dataset, demonstrates the capability to accomplish unseen downstream tasks through human instructions.

$$v_d^{\alpha_i} = \lfloor \text{clip}(v^{\alpha_i}, 0, 1-\delta) \cdot V \rfloor + (i-1)V, \;\; i=1,\cdots,k \qquad (3)$$

where $\delta$ is a small slack variable. This ensures that each of the $V$ possible cases for each attribute is assigned a unique token. As a result, the embedding layer outputs a vector that contains information about both attribute category and strength. This vector is then multiplied with the mask, passed through a multi-head self-attention layer and a linear layer, and used as a conditioning input to the noise predictor. Furthermore, due to our requirement of a large receptive field to capture the attributes on the trajectory level, we employ a transformer-based backbone, DiT (Peebles & Xie, 2023), for our noise predictor, instead of the commonly used UNet (Ronneberger et al., 2015). To adopt DiT for understanding relative attribute strengths and guiding decision trajectory generation, we made several structural modifications. (See Fig. 2, part AlignDiff Architecture). Combining all the above components, we train a diffusion model with the noise predictor loss, where $\boldsymbol{m}^{\boldsymbol{\alpha}}$ is sampled from a binomial distribution $\mathcal{B}(k, p)$, and $p$ represents the no-masking probability.

$$\mathcal{L}(\phi) = \mathbb{E}_{(\boldsymbol{x}_0,\boldsymbol{v}^{\boldsymbol{\alpha}})\sim\mathcal{D}_G, t\sim\text{Uniform}(T), \epsilon\sim\mathcal{N}(\boldsymbol{0},\boldsymbol{I}), \boldsymbol{m}^{\boldsymbol{\alpha}}\sim\mathcal{B}(k,p)}||\epsilon - \epsilon_\phi(\boldsymbol{x}_t, t, \boldsymbol{v}^{\boldsymbol{\alpha}}, \boldsymbol{m}^{\boldsymbol{\alpha}})||_2^2 \qquad (4)$$

## 4.4 ALIGNDIFF INFERENCE

With an attribute strength model $\hat{\zeta}_\theta^{\boldsymbol{\alpha}}$ and a noise predictor $\epsilon_\phi$, we can proceed to plan with AlignDiff. Suppose at state $s_t$, given a preference $(\boldsymbol{v}^{\boldsymbol{\alpha}}, \boldsymbol{m}^{\boldsymbol{\alpha}})$, we use DDIM sampler with a subsequence $\kappa$ of

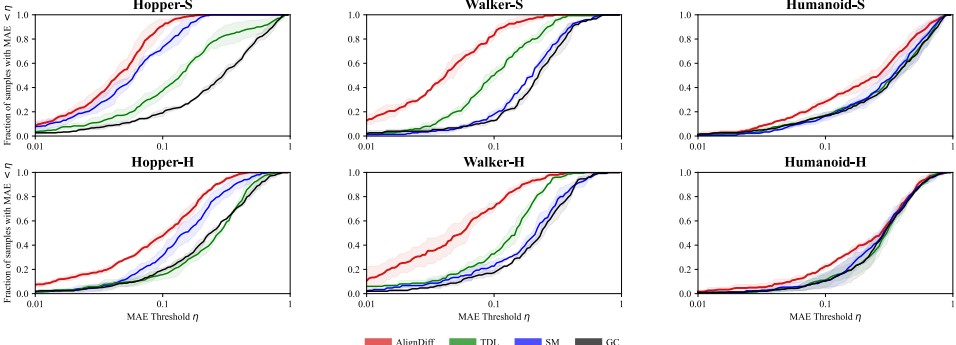

Figure 4: MAE curves. The vertical axis represents the MAE threshold, which is presented using a logarithmic scale. The horizontal axis represents the percentage of samples below the MAE threshold. Each point $(x, y)$ on the curve indicates that the algorithm has a probability of $y$ to achieve an MAE between the agent's relative attribute strength and the desired value below $x$. A larger area enclosed by the curve and the axes indicate better performance in matching human preferences.

length $S$ to iteratively generate candidate trajectories.

$$\boldsymbol{x}_{\kappa_{i-1}} = \sqrt{\xi_{\kappa_{i-1}}}\big(\frac{\boldsymbol{x}_{\kappa_i} - \sqrt{1 - \xi_{\kappa_i}}\tilde{\epsilon}_\phi(\boldsymbol{x}_{\kappa_i})}{\sqrt{\xi_{\kappa_i}}}\big) + \sqrt{1 - \xi_{\kappa_{i-1}} - \sigma_{\kappa_i}^2}\tilde{\epsilon}_\phi(\boldsymbol{x}_{\kappa_i}) + \sigma_{\kappa_i}\epsilon_{\kappa_i} \qquad (5)$$

During the process, we fix the current state $s_t$ as it is always known. Each $\tau$ in the candidate trajectories satisfies human preference $(\boldsymbol{v}^{\boldsymbol{\alpha}}, \boldsymbol{m}^{\boldsymbol{\alpha}})$ a priori. Then we utilize $\hat{\zeta}_\theta^{\boldsymbol{\alpha}}$ to criticize and select the most aligned one to maximize the following objective:

$$\mathcal{J}(\tau) = ||(\boldsymbol{v}^{\boldsymbol{\alpha}} - \hat{\zeta}_\theta^{\boldsymbol{\alpha}}(\tau)) \circ \boldsymbol{m}^{\boldsymbol{\alpha}}||_2^2 \qquad (6)$$

The first step of the chosen plan is executed in the environment. For the convenience of readers, we summarize the training and inference phases of AlignDiff in Algorithm 1 and Algorithm 2. Additionally, to make interaction easier, we add a natural language control interface. By keeping an instruction corpus and using Sentence-BERT (Reimers & Gurevych, 2019) to calculate the similarity between the instruction and the sentences in the corpus, we can find the intent that matches the closest and modify the attributes accordingly.

## 5 EXPERIMENTS

We conduct experiments on various locomotion tasks from MuJoCo (Todorov et al., 2012) and DMControl (Tunyasuvunakool et al., 2020a) to evaluate the preference aligning capability of the algorithm. Through the experiments, we aim to answer the following research questions (RQs):
**Matching (RQ1):** Can AlignDiff better align with human preferences compared to other baselines?
**Switching (RQ2):** Can AlignDiff quickly and accurately switch between different behaviors?
**Covering (RQ3):** Can AlignDiff cover diverse behavioral distributions in the dataset?
**Robustness (RQ4):** Can AlignDiff exhibit robustness to noisy datasets and limited feedback?

### 5.1 EXPERIMENTAL SETUP

**Benchmarks:** We select hopper, walker, and humanoid locomotion tasks as the benchmarks (See Appendix A for more details). Predefined attributes are presented in Table 1. In each experiment, we compare **s**ynthetic labels generated by scripts (denote as **S**) and **h**uman labels collected by crowdsourcing (denote as **H**) separately to demonstrate the generalizability of AlignDiff.

**Baselines:** There are several existing paradigms for constructing policies conditioned on human preferences, which we use as baselines in our experiments (See Appendix C for more details):

Table 1: Predefined attributes for each task.

| Environment | Attributes |
|---|---|
| Hopper | Speed
Jump height |
| Walker | Speed
Torso height
Stride length
Left-right leg preference
Humanness |
| Humanoid | Speed
Head height
Humanness |

Table 2: Area enclosed by the MAE curve. A larger value indicates better alignment performance. Performance on the humanness attribute is reported in a separate row.

| Type/Attribute | Environment | GC(Goal conditioned BC) | SM(Sequence Modeling) | TDL(TD Learning) | AlignDiff |
|---|---|---|---|---|---|
| Synthetic | Hopper | $0.285 \pm 0.009$ | $0.573 \pm 0.008$ | $0.408 \pm 0.030$ | $\mathbf{0.628 \pm 0.026}$ |
| | Walker | $0.319 \pm 0.005$ | $0.333 \pm 0.007$ | $0.445 \pm 0.012$ | $\mathbf{0.621 \pm 0.023}$ |
| | Humanoid | $0.252 \pm 0.016$ | $0.258 \pm 0.030$ | $0.258 \pm 0.004$ | $\mathbf{0.327 \pm 0.016}$ |
| Average | | 0.285 | 0.388 | 0.370 | **0.525** |
| Human | Hopper | $0.305 \pm 0.013$ | $0.387 \pm 0.031$ | $0.300 \pm 0.024$ | $\mathbf{0.485 \pm 0.013}$ |
| | Walker | $0.322 \pm 0.011$ | $0.341 \pm 0.009$ | $0.435 \pm 0.005$ | $\mathbf{0.597 \pm 0.018}$ |
| | Humanoid | $0.262 \pm 0.026$ | $0.272 \pm 0.029$ | $0.262 \pm 0.027$ | $\mathbf{0.313 \pm 0.015}$ |
| Average | | 0.296 | 0.333 | 0.332 | **0.465** |
| Humanness | Walker | $0.334 \pm 0.035$ | $0.448 \pm 0.003$ | $0.510 \pm 0.006$ | $\mathbf{0.615 \pm 0.022}$ |
| | Humanoid | $0.285 \pm 0.020$ | $0.320 \pm 0.021$ | $0.257 \pm 0.003$ | $\mathbf{0.396 \pm 0.020}$ |
| Average | | 0.310 | 0.384 | 0.384 | **0.510** |

• **Goal conditioned behavior clone (GC)** leverages supervised learning to train a goal conditioned policy $\pi(s|v^{\alpha}, m^{\alpha})$ for imitating behaviors that match human preferences. We implement this baseline following RvS (Emmons et al., 2022).

• **Sequence modeling (SM)** is capable of predicting the optimal action based on historical data and prompt tokens, allowing it to leverage the simplicity and scalability of Transformer architectures. We adopt the structure of Decision Transformer (DT) (Chen et al., 2021) and incorporate the preference $(v^{\alpha}, m^{\alpha})$ as an additional token in the input sequence.

• **TD Learning (TDL)** is a classic RL paradigm for learning optimal policies. TD3BC (Fujimoto & Gu, 2021) is one such algorithm designed for offline RL settings. Since a reward model is required to provide supervised training signals for policy optimization, we distill an attribute-conditioned reward model from $\hat{\zeta}_{\theta}^{\alpha}$ to train TD3BC. This serves as an improved version of RBA.

Throughout the experiments, all algorithms share the same attribute strength model to ensure fairness. The source of human feedback datasets can be found in Appendix B.

## 5.2 MATCHING (RQ1)

**Evaluation by attribute strength model:** We conducted multiple trials to collect the mean absolute error (MAE) between the evaluated and target relative strengths. For each trial, we sample an initial state $s_0$, a target strengths $v_{\text{targ}}^{\alpha}$, and a mask $m^{\alpha}$, as conditions for the execution of each algorithm. Subsequently, the algorithm runs for $T$ steps, resulting in the exhibited relative strengths $v^{\alpha}$ evaluated by $\hat{\zeta}_{\theta}$. We then calculated the percentage of samples that fell below pre-designed thresholds to create the MAE curves presented in Fig. 4. The area enclosed by the curve and the axes were used to define a metric, which is presented in Table 2. A larger metric value indicates better performance in matching. Experiments show that AlignDiff performs significantly better than other baselines on the Hopper and Walker benchmarks. On the Humanoid benchmark, AlignDiff is the only one that demonstrates preferences aligning capability, while the other baselines fail to learn useful policies. We also find that AlignDiff exhibits slightly better performance on synthetic labels, which may be attributed to the script evaluation being inherently rational without any noise.

**Evaluation by humans** We further conducted a questionnaire-based evaluation. Specifically, we instructed the algorithm to adjust an attribute to three different levels (corresponding to 0.1, 0.5, and 0.9), resulting in three video segments. The order of the videos was shuffled, and human evaluators were asked to sort them. A total of 2,160 questionnaires were collected from 424 human evaluators. The sorting accuracy results are reported in Table 3, in which a higher accuracy indicates better performance in matching human preferences. We observe that AlignDiff performs significantly better than other baselines across different environments. This leads us to conclude that AlignDiff successfully utilizes the powerful conditional generation abilities of diffusion models to exhibit notable differences in the specified attributes. Additionally, we find that sorting accuracy is much higher when using human labels compared to synthetic labels. This suggests that human labels provide a level of intuition that synthetic labels cannot, resulting in better alignment with human preferences. To ensure impartiality, we have also conducted a t-test on the evaluator groups. We refer to Appendix D for more information on the evaluation process.

Table 3: Accuracy of videos sorting by human evaluators. A higher accuracy indicates better performance in matching human preferences. Performance on the humanness attribute is reported in a separate row.

| Label/Attribute | Environment | **GC**(Goal conditioned BC) | **SM**(Sequence Modeling) | **TDL**(TD Learning) | **AlignDiff** |
|---|---|---|---|---|---|
| Synthetic | Hopper | 0.67 | 80.86 | 36.23 | **85.45** |
| | Walker | 5.86 | 9.72 | 49.49 | **70.23** |
| | Humanoid | 19.88 | 19.88 | 4.02 | **65.97** |
| Average | | 8.80 | 38.47 | 29.91 | **73.88** |
| Human | Hopper | 0.00 | 32.78 | 15.56 | **95.24** |
| | Walker | 4.81 | 2.55 | 59.00 | **93.75** |
| | Humanoid | 21.82 | 27.66 | 0.00 | **66.10** |
| Average | | 7.06 | 23.12 | 24.85 | **85.03** |
| Humanness | Walker | 2.88 | 1.02 | 64.00 | **93.33** |
| | Humanoid | 21.82 | 27.66 | 0.00 | **72.88** |
| Average | | 12.35 | 14.34 | 32.00 | **83.11** |

**Investigation of the humanness attribute:** We conducted a specific human evaluation to assess the humanness attribute, and the results are reported in a separate row of both Table 2 and Table 3. In terms of humanness, we observe that AlignDiff performs significantly better than other baselines. With the aid of human labels, AlignDiff is able to effectively capture the features of human motion patterns and exhibit the most human-like behavior.

### 5.3 SWITCHING (RQ2)

**Track the changing target attributes:** To evaluate the ability of the learned model to switch between different behaviors, we conducted an attribute-tracking experiment on the Walker benchmark. Starting from the same initial state, we ran each algorithm for 800 steps, modifying the target attributes $v_{\text{targ}}^{\alpha}$ at steps $(0, 200, 400, 600)$, and recorded the actual speed and torso height of the robot The tracking curves are presented in Fig. 6. We observe that AlignDiff quickly and accurately tracked the ground truth, whereas the other baselines showed deviations from it, despite demonstrating a trend in attribute changes.

**Complete unseen tasks by attribute instructions:** In addition, we tested AlignDiff's zero-shot capability under human instructions by deploying it to unseen downstream tasks. By adjusting attributes such as speed, torso height, and stride length by the human instructor, the walker robot, which was only trained on locomotion datasets, successfully completed the gap-crossing and obstacle avoidance tasks from Bisk benchmark (Gehring et al., 2021). Fig. 3 presents selected key segments of this test, highlighting the promising potential of AlignDiff for human-AI collaboration.

### 5.4 COVERING (RQ3)

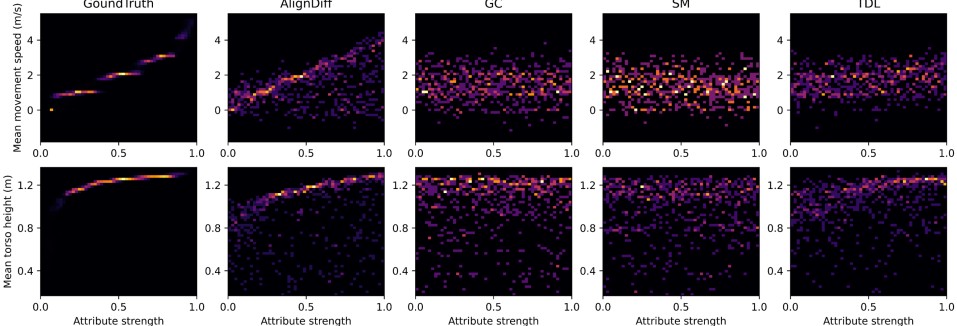

Figure 5: Distribution plots of attribute strength values and actual attributes are shown. The horizontal axis represents the attribute strength value $v$, while the vertical axis represents the corresponding actual attribute $u$. $p(u, v)$ denotes the probability that, given a target attribute strength value $v$, the algorithm can produce a trajectory with the actual attribute $u$. The color gradient in the distribution plot represents the probability $p(u, v)$, ranging from dark to light.

To align with complex and variable human intention, AlignDiff requires the capability to cover a diverse range of behaviors within the offline datasets. This investigation aims to determine: *Can*

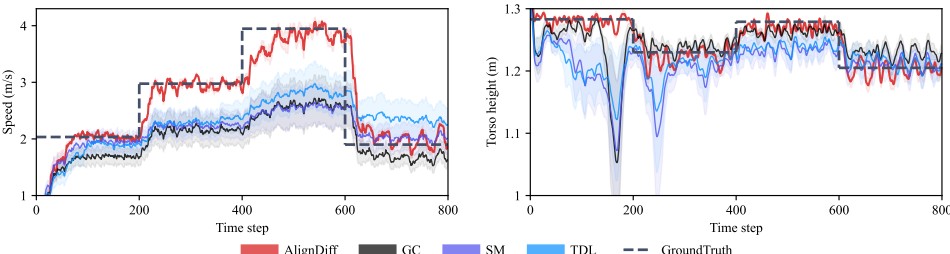

Figure 6: Visualization of behavior switching. The target attributes of speed and torso height at steps 0, 200, 400, and 600 are set to $[0.5, 0.75, 0.875, 0.3875]$ and $[0.8, 0.4, 0.7, 0.35]$, respectively. We obtain the actual attributes corresponding to the target from the dataset distribution in Fig. 5 as the ground truth.

Table 4: Area enclosed by the MAE curve of AlignDiff trained on noisy datasets.

| Environment | **GC**(Goal conditioned BC) | **SM**(Sequence Modeling) | **TDL**(TD Learning) | **AlignDiff** |
|---|---|---|---|---|
| oracle | $0.319 \pm 0.007$ | $0.338 \pm 0.008$ | $0.455 \pm 0.012$ | $\mathbf{0.634 \pm 0.020}$ |
| 20% noise | $0.320 \pm 0.014$ | $0.327 \pm 0.013$ | $0.323 \pm 0.031$ | $\mathbf{0.559 \pm 0.041}$ |
| 50% noise | $0.300 \pm 0.022$ | $0.281 \pm 0.014$ | $0.305 \pm 0.028$ | $\mathbf{0.384 \pm 0.022}$ |

*AlignDiff produce the most probable dynamics within the datasets, and even combine different attributes to produce unseen behaviors?* We compare the distribution of $p(u, v)$, which represents the likelihood that the algorithm produces trajectories with the actual attribute $u$ given the target strength $v$, between the algorithm and the datasets. Due to intractability, Monte Carlo methods are used to approximate it, as detailed in Appendix E. We focus on the speed and torso height attributes defined in the Walker benchmark, as the corresponding physical quantities can be directly obtained from the MuJoCo engine. As shown in Fig. 5, we report the distributions corresponding to each algorithm and observe that AlignDiff not only covers the behaviors in the datasets but also fills in the "disconnected regions" in the ground truth distribution, indicating the production of unseen behaviors.

## 5.5 ROBUSTNESS (RQ4)

We evaluate the robustness of our algorithm from two aspects: 1) **Robustness to dataset noise.** We established two additional datasets for the Walker benchmark, where random decision trajectories are mixed into the original datasets at proportions of 20% and 50%, respectively. We train algorithms on the original datasets and these two noisy datasets. The performance is shown in Table 4. The

Table 5: Performance of AlignDiff trained with a different number of feedback labels.

| Number of labels | Area |
|---|---|
| 10,000 | $0.628 \pm 0.026$ |
| 2,000 (80%↓) | $0.621 \pm 0.013$ (1.11%↓) |
| 500 (95%↓) | $0.526 \pm 0.029$ (16.2%↓) |

experiment shows that AlignDiff has the best robustness to noise. 2) **Robustness to the number of feedback labels.** We compare the performance of AlignDiff trained on 10k, 2k, and 500 synthetic labels on the Hopper benchmark, as shown in Table 5. The experiment shows that the performance does not decrease significantly when the number of feedback labels decreases from 10k to 2k, and only some performance loss is observed when the number decreases to 500. This result indicates that AlignDiff can achieve good performance with fewer feedback labels.

## 6 CONCLUSION

In this paper, we introduce AlignDiff, a novel framework that achieves zero-shot human preference aligning. Our framework consists of two main components. The first component utilizes RLHF technology to quantify human preferences, addressing the *abstractness* of human preferences. The second component includes a behavior-customizable diffusion model, which can plan to accurately match desired behaviors and efficiently switch from one to another, addressing the *mutability* of human preferences. We conducted various experiments to evaluate the algorithm's capability of preference matching, switching, and covering. The results demonstrate that AlignDiff outperforms other strong baselines with exceptional performance. However, like other diffusion-based RL algorithms (Hegde et al., 2023), AlignDiff is slow at inference time due to the iterative sampling process. Performing faster sampling may mitigate the issue. The ability of AlignDiff to accomplish unseen tasks under human instructions showcases its potential to combine skills and complete complex tasks under the command of higher-level models (e.g. LLMs), which we leave as future work.

ACKNOWLEDGMENTS

This work is supported by the National Key R&D Program of China (Grant No. 2022ZD0116402), the National Natural Science Foundation of China (Grant Nos. 92370132) and the Xiaomi Young Talents Program of Xiaomi Foundation. We thank YouLing Crowdsourcing for their assistance in annotating the dataset.

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

# A    TASKS AND DATASETS

In this section, we will provide a detailed description of three locomotion tasks that serve as experimental benchmarks, along with an explanation of how we collected the corresponding offline datasets.

**Hopper.** Hopper is a single-legged robot selected from the Gym-MuJoCo locomotion tasks. We use pre-trained PPO (Schulman et al., 2017) agents provided by PEDA (Zhu et al., 2023) to collect a total of 5 million time steps for our datasets. The relative magnitudes of speed and height provided by PEDA are used to generate synthetic labels.

**Walker.** Walker is a simplified humanoid bipedal robot selected from the deepmind control suite benchmarks (Tunyasuvunakool et al., 2020b). To collect an offline dataset, we modify the reward function of the original Walker benchmark and randomly select target velocities and heights. We train 32 policies with SAC (Haarnoja et al., 2018). These policies are used to collect a total of 3.2 million time steps for our datasets. To generate synthetic labels, we retrieve the corresponding physical quantities from the MuJoCo engine to compute and compare the attributes of two trajectories. However, humanness can only be obtained through human feedback, making it infeasible to employ in the synthetic labels setting.

**Humanoid.** Humanoid is a 3D bipedal robot designed to simulate a human. To collect an offline dataset, we modify the reward function of the original Humanoid benchmark and randomly select target velocities and heights. We train 40 policies with SAC. These policies are used to collect a total of 4 million time steps for our datasets. To generate synthetic labels, we retrieve the corresponding physical quantities from the MuJoCo engine to compute and compare the attributes of two trajectories. However, humanness can only be obtained through human feedback, making it infeasible to employ in the synthetic labels setting.

# B    HUMAN FEEDBACK COLLECTION DETAILS

Human preferences are influenced by various internal and external factors (Wang et al., 2017; 2022; 2018; 2020). Hence, in this section, we provide our human feedback collection details. We recruited a total of 100 crowdsourcing workers to annotate 4,000 feedback labels for each environment. Each worker was assigned to annotate 120 labels. They were given video pairs that lasted for about 3 seconds (100-time steps, generating videos at 30 frames per second) and were asked to indicate which showed stronger performance based on pre-defined attributes. We made sure that each worker was compensated fairly and provided them with a platform that allowed them to save their progress and stop working at any time. There is no reason to believe that crowdsourcing workers experienced any physical or mental risks in the course of these studies. The task description provided to crowdsourcing workers is as follows:

---

**Hopper**

**Question:** For each attribute, which side of the video pair shows a stronger performance?
**Options:** `(Left Side, Equal, Right Side)`
**Definitions of pre-defined attributes:**

- **Speed**: The speed at which the agent moves to the right. The greater the distance moved to the right, the faster the speed.

- **Jump height**: If the maximum height the agent can reach when jumping is higher, stronger performance should be selected. Conversely, weaker performance should be selected if the maximum height is lower. If it is difficult to discern which is higher, select equal.

---

**Walker**

**Question:** For each attribute, which side of the video pair shows a stronger performance?
**Options:** `(Left Side, Equal, Right Side)`
**Definitions of pre-defined attributes:**

---

- **Speed**: The speed at which the agent moves to the right. The greater the distance moved to the right, the faster the speed.
- **Stride**: The maximum distance between the agent's feet. When the agent exhibits abnormal behaviors such as falling, shaking, or unstable standing, weaker performance should be selected.
- **Leg Preference**: If the agent prefers the left leg and exhibits a walking pattern where the left leg drags the right leg, a stronger performance should be selected. Conversely, weaker performance should be selected. If the agent walks with both legs in a normal manner, equal should be selected.
- **Torso height**: The torso height of the agent. If the average height of the agent's torso during movement is higher, stronger performance should be selected. Conversely, weaker performance should be selected if the average height is lower. If it is difficult to discern which is higher, select equal.
- **Humanness**: The similarity between agent behavior and humans. When the agent is closer to human movement, stronger performance should be selected. When the agent exhibits abnormal behaviors such as falling, shaking, or unstable standing, weaker performance should be selected.

---

### Humanoid

**Question:** For each attribute, which side of the video pair shows a stronger performance?
**Options:** (Left Side, Equal, Right Side)
**Definitions of pre-defined attributes:**

- **Speed**: The speed at which the agent moves to the right. The greater the distance moved to the right, the faster the speed.
- **Head height**: The head height of the agent. If the average height of the agent's head during movement is higher, stronger performance should be selected. Conversely, weaker performance should be selected if the average height is lower. If it is difficult to discern which is higher, select equal.
- **Humanness**: The similarity between agent behavior and humans. When the agent is closer to human movement, stronger performance should be selected. When the agent exhibits abnormal behaviors such as falling, shaking, or unstable standing, weaker performance should be selected.

## C  BASELINES DETAILS

In this section, we introduce the implementation details of each baseline in our experiments, the reasonable and necessary modifications compared to the original algorithm, and the reasons for the modifications.

Table 6: Hyperparameters of GC(Goal conditioned behavior clone).

| Hyperparameter | Value |
|---|---|
| Hidden layers | 2 |
| Layer width | 1024 |
| Nonlinearity | ReLU |
| Batch size | 32 |
| Optimizer | Adam |
| Learning rate | $10^{-3}$ |
| Gradient steps | $5 \times 10^5$ Hopper/Walker $10^6$ Humanoid |

Table 7: Hyperparameters of SM(Sequence modeling).

| Hyperparameter | Value |
|---|---|
| Number of layers | 3 |
| Number of attention heads | 1 |
| Embedding dimension | 128 |
| Nonlinearity | ReLU |
| Batch size | 32 |
| Optimizer | Adam |
| Learning rate | $10^{-4}$ |
| Weight decay | $10^{-4}$ |
| Dropout | 0.1 |
| Context length | 32 Walker
100 Hopper/Humanoid |
| Gradient steps | $5 \times 10^5$ Hopper/Walker
$10^6$ Humanoid |

Table 8: Hyperparameters of distilled reward model.

| Hyperparameter | Value |
|---|---|
| Hidden layers | 2 |
| Layer width | 512 |
| Nonlinearity | ReLU |
| Batch size | 256 |
| Optimizer | Adam |
| Learning rate | $10^{-4}$ |
| Length of trajectories | 100 |
| Gradient steps | 5000 |

## C.1 GC(GOAL CONDITIONED BEHAVIOR CLONE)

GC leverages supervised learning to train a goal conditioned policy $\pi_\theta(a_t|s_t, \boldsymbol{v^\alpha}, \boldsymbol{m^\alpha})$. We implement this baseline based on RvS (Emmons et al., 2022) and choose human preferences $(\boldsymbol{v^\alpha}, \boldsymbol{m^\alpha})$ as $w$. Following the best practice introduced in RvS, we implement GC policy as a 3-layer MLP, which formulates a truncated Gaussian distribution. Given dataset $\mathcal{D}_G = \{\tau^H, \boldsymbol{v^\alpha}\}$, we optimize the policy to maximize:

$$\sum_{(\tau^H, \boldsymbol{v^\alpha}) \in \mathcal{D}_G} \sum_{1 \le t \le H} \mathbb{E}_{\boldsymbol{m^\alpha} \sim \mathcal{B}(k,p)}[\log \pi_\theta(a_t|s_t, \boldsymbol{v^\alpha}, \boldsymbol{m^\alpha})] \tag{7}$$

The hyperparameters are presented in Table 6. The selection is nearly identical to the choices provided in Emmons et al. (2022).

## C.2 SM(SEQUENCE MODELING)

SM is capable of predicting the optimal action based on historical data and prompt tokens, allowing it to leverage the simplicity and scalability of Transformer architectures. Following the structure introduced in DT (Chen et al., 2021), we design the structure of the SM baseline as a causal Transformer. In comparison to DT, where reward is present as a signal, our task only involves a target strength vector $\boldsymbol{v^\alpha}$ and an attribute mask $\boldsymbol{m^\alpha}$. Therefore, we remove the RTG tokens and instead set the first token as the embedding of $(\boldsymbol{v^\alpha}, \boldsymbol{m^\alpha})$. During the training phase, SM is queried to predict the current action $a_t$ based on $(\boldsymbol{v^\alpha}, \boldsymbol{m^\alpha})$, historical trajectories $(s_{<t}, a_{<t})$, and the current state $s_t$. We optimize SM to minimize:

$$\sum_{(\tau^H, \boldsymbol{v^\alpha}) \in \mathcal{D}_G} \sum_{1 \le t \le H} \mathbb{E}_{\boldsymbol{m^\alpha} \sim \mathcal{B}(k,p)}[||f(s_{\le t}, a_{<t}, \boldsymbol{v^\alpha}, \boldsymbol{m^\alpha}) - a_t||_2^2] \tag{8}$$

Table 9: Hyperparameters of TDL(TD learning).

| Hyperparameter | Value |
|---|---|
| Hidden layers | 2 |
| Layer width | 512 |
| Number of Q functions | 2 |
| Nonlinearity | ReLU |
| Discount | 0.99 |
| Tau | 0.005 |
| Policy noise | 0.2 |
| Noise clip | 0.5 |
| Policy frequency | 2 |
| Alpha | 2.5 |
| Batch size | 32 |
| Optimizer | Adam |
| Learning rate | $3 \times 10^{-4}$ |
| Gradient steps | $5 \times 10^5$ Hopper/Walker $10^6$ Humanoid |

During the inference phase, we use $a_t = f(s_{\leq t}, a_{<t}, \boldsymbol{v^\alpha}, \boldsymbol{m^\alpha})$ as our policy and continuously store state-action pair into the historical trajectory. The hyperparameters are presented in Table 7. The selection is nearly identical to the choices provided in Chen et al. (2021).

## C.3 TDL(TD LEARNING)

TD learning is a classic RL paradigm for learning optimal policies. Since a reward model is needed to provide supervised training signals, we first need to distill $\hat{\zeta}_\theta^{\boldsymbol{\alpha}}$ into a reward model. Following the approach in RBA, we train an attribute-strength-conditioned reward model, denoted as $r_\theta(s_t, a_t, \boldsymbol{v^\alpha}, \boldsymbol{m^\alpha})$. We begin by using the reward model to define an attribute proximity probability:

$$P[\tau_1 \succ \tau_2 | \boldsymbol{v^\alpha}, \boldsymbol{m^\alpha}] = \frac{\exp \sum_t r_\theta(s_t^1, a_t^1, \boldsymbol{v^\alpha}, \boldsymbol{m^\alpha})}{\sum_{i \in \{1,2\}} \exp \sum_t r_\theta(s_t^i, a_t^i, \boldsymbol{v^\alpha}, \boldsymbol{m^\alpha})} \quad (9)$$

This equation represents the probability that attribute strength of $\tau_1$ is more aligned with $(\boldsymbol{v^\alpha}, \boldsymbol{m^\alpha})$ compared to $\tau_2$. Subsequently, we obtain pseudo-labels through the attribute strength model:

$$y(\tau_1, \tau_2, \boldsymbol{v^\alpha}) = \begin{cases} (1, 0), & \text{if } ||(\hat{\zeta}^{\boldsymbol{\alpha}}(\tau_1) - \boldsymbol{v^\alpha}) \circ \boldsymbol{m^\alpha}||_2 \leq ||(\hat{\zeta}^{\boldsymbol{\alpha}}(\tau_2) - \boldsymbol{v^\alpha}) \circ \boldsymbol{m^\alpha}||_2 \\ (0, 1), & \text{if } ||(\hat{\zeta}^{\boldsymbol{\alpha}}(\tau_1) - \boldsymbol{v^\alpha}) \circ \boldsymbol{m^\alpha}||_2 > ||(\hat{\zeta}^{\boldsymbol{\alpha}}(\tau_2) - \boldsymbol{v^\alpha}) \circ \boldsymbol{m^\alpha}||_2 \end{cases} \quad (10)$$

With these pseudo-labels, we train the distilled reward model by optimizing a cross-entropy loss:

$$-\mathbb{E}_{\boldsymbol{v^\alpha}} \mathbb{E}_{\boldsymbol{m^\alpha} \sim \mathcal{B}(k,p)} \mathbb{E}_{(\tau_1, \tau_2)} y(1) \log P[\tau_1 \succ \tau_2 | \boldsymbol{v^\alpha}, \boldsymbol{m^\alpha}] + y(2) P[\tau_2 \succ \tau_1 | \boldsymbol{v^\alpha}, \boldsymbol{m^\alpha}] \quad (11)$$

Intuitively, if executing action $a_t$ under state $s_t$ leads to a trajectory that exhibits attribute strength closer to $(\boldsymbol{v^\alpha}, \boldsymbol{m^\alpha})$, the corresponding reward $r_\theta(s_t, a_t, \boldsymbol{v^\alpha}, \boldsymbol{m^\alpha})$ will be larger; conversely, if the attribute strength is farther from $(\boldsymbol{v^\alpha}, \boldsymbol{m^\alpha})$, the reward will be smaller.

Since the reward model now conditions the attribute strength, we need to define the attribute-strength-conditioned Q function $Q(s, a, \boldsymbol{v^\alpha}, \boldsymbol{m^\alpha})$ and the policy $\pi(s, \boldsymbol{v^\alpha}, \boldsymbol{m^\alpha})$. With all these components, we can establish a TDL baseline on top of TD3BC and optimize the policy by maximizing:

$$\sum_{(\tau^H, \boldsymbol{v^\alpha}) \in \mathcal{D}_G} \sum_{1 \leq t \leq H} \mathbb{E}_{\boldsymbol{m^\alpha} \sim \mathcal{B}(k,p)} [\lambda Q(s_t, \pi(s_t, \boldsymbol{v^\alpha}, \boldsymbol{m^\alpha}), \boldsymbol{v^\alpha}, \boldsymbol{m^\alpha}) - (\pi(s, \boldsymbol{v^\alpha}, \boldsymbol{m^\alpha}) - a)^2] \quad (12)$$

The hyperparameters are presented in Table 8 and Table 9. The selection is nearly identical to the choices provided in Fujimoto & Gu (2021).

## D HUMAN EVALUATION DETAILS

### D.1 EVALUATION PROCESS

Human evaluation is conducted in the form of questionnaires, which include multiple attribute strength ranking questions. The flow chart of the human evaluation process is summarized in Fig. 7. We collected a total of 270 sets of videos for each algorithm, resulting in a total of 2,160 questionnaires. And we invite a total of 424 human evaluators to participate in the experiment. Before the questionnaire, each evaluator is asked to provide basic demographic information to help us understand the distribution of evaluators. Demographic

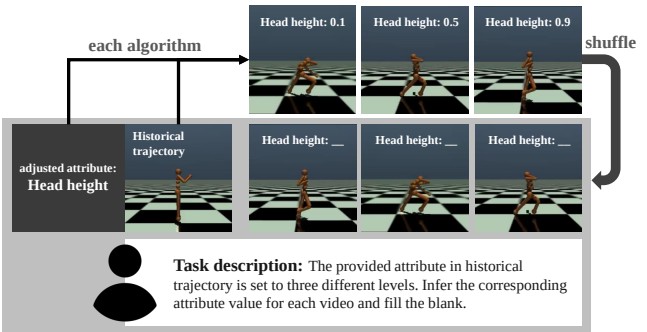

Figure 7: The flowchart of the human evaluation process introduced in Section 5.2. The information visible to human assessors is enclosed within gray boxes.

information included gender, age, education, and experience with AI. The distribution of evaluators for each task is presented in Fig. 8. In each questionnaire, in addition to the relevant videos mentioned in Section 5.2, we provide evaluators with a task description of the environment and detailed explanations of the attributes being evaluated. The textual descriptions are as follows:

---

### Hopper

**Description:** Hopper is a single-legged robot that adjusts its movement based on predefined attributes. Each attribute, such as `Movement Speed`, ranges from 0 to 1, with higher values indicating stronger attributes (e.g., a value of 1 represents the fastest speed). In each task, we will set one attribute to three different levels: 0, 1, and 2 (corresponding to attribute strengths of 0.1, 0.5, and 0.9 respectively) for 5 seconds of continued movement, in random order. You will see 3 videos with modified attributes. Your task is to infer the corresponding attribute value for each video. For example, if a video exhibits the strongest attribute performance, you should select 2 from the options below. Each option can only be chosen once. If you find it challenging to determine an attribute for a particular video, you can mark it as `None`. The `None` option can be selected multiple times.

**Attribute explanation:**

- Movement Speed: The speed of movement, with higher attribute values indicating faster speed. Generally, the faster the movement of the floor grid texture, the faster the speed.

- Jump Height: The maximum height that can be reached by jumping, with higher attribute values corresponding to higher jump heights. Jump height can be determined by the jump posture or the size of the texture of the floor grid.

---

### Walker

**Description:** Walker is a bipedal robot that adjusts its movement based on predefined attributes. Each attribute, such as `Movement Speed`, ranges from 0 to 1, with higher values indicating stronger attributes (e.g., a value of 1 represents the fastest speed). In each task, you'll be shown a 3-second video of Walker's past movements and informed about an attribute along with its current strength value. Next, we will set the attribute to three different levels: 0, 1, and 2 (corresponding to attribute strengths of 0.1, 0.5, and 0.9 respectively) for the following 5 seconds of continued movement, in random order. You will see 3 videos with modified attributes. Your task is to infer the corresponding attribute value for each video. For example, if a video exhibits the strongest attribute performance, you should select 2 from the options below. Each option can only be chosen once. If you find it challenging to determine an attribute for a particular video, you can mark it as `None`. The `None` option can be selected multiple times.

**Attribute explanation:**

- Movement Speed: The speed of movement, with higher attribute values indicating faster speed. Generally, the faster the movement of the floor grid texture, the faster the speed.

- Stride Length: The maximum distance between the feet, with higher attribute values corresponding to larger strides. Stride length can be determined by the number of grid squares crossed in a single step.

- Left-Right Leg Preference: A higher attribute value indicates Walker's tendency to use the left leg to initiate right leg movement; a lower value indicates a preference for the right leg initiating left leg movement; when the attribute value is close to 0.5, Walker tends to use both legs simultaneously.

- Torso Height: The height of the torso, with higher attribute values indicating greater torso height. Generally, the smaller the angle between the legs, the higher the torso; the larger the angle between the legs, the lower the torso.

- Humanness: The degree of similarity between Walker's behaviors and human behaviors, with higher attribute values indicating greater similarity.

---

### Humanoid

**Description:** Humanoid is a bipedal robot that adjusts its movement based on predefined attributes. Each attribute, such as `Movement Speed`, ranges from 0 to 1, with higher values indicating stronger attributes (e.g., a value of 1 represents the fastest speed). In each task, we will set one attribute to three different levels: 0, 1, and 2 (corresponding to attribute strengths of 0.1, 0.5, and 0.9 respectively) for 5 seconds of continued movement, in random order. You will see 3 videos with modified attributes. Your task is to infer the corresponding attribute value for each video. For example, if a video exhibits the strongest attribute performance, you should select 2 from the options below. Each option can only be chosen once. If you find it challenging to determine an attribute for a particular video, you can mark it as `None`. The `None` option can be selected multiple times.

**Attribute explanation:**

- Movement Speed: The speed of movement, with higher attribute values indicating faster speed. Generally, the faster the movement of the floor grid texture, the faster the speed.

- Head Height: The height of the head, with higher attribute values indicating greater head height.

- Humanness: The degree of similarity between Humanoid's behaviors and human behaviors, with higher attribute values indicating greater similarity.

---

### D.2 T-TEST FOR HUMAN EVALUATORS

Table 10: T-test result for human evaluation.

| Classification | t-statistic | p-value |
|---|---|---|
| Male/Female | 0.2231 | 0.8234 |
| No experience/Experienced | 0.1435 | 0.8859 |

To ensure the reliability of our designed human evaluation, we conducted a t-test on the group of evaluators to analyze whether there was any bias in the questionnaire. We divide the participants into two groups based on gender and whether they have AI learning experience, respectively. We set the hypothesis that *there is no significant difference in the average accuracy of the questionnaire evaluation between the two groups*. If the hypothesis is accepted, we can conclude that the questionnaire design is unbiased and the experiment is reliable. If the hypothesis is rejected, we conclude that there is bias in the questionnaire design, which may cause certain groups to make biased judgments, rendering the experimental design unreliable. The results of the t-test for the two groups under the two classification methods are presented in Table 10. The results show that the p-values are higher than the significance level (0.05). Therefore, we can accept the null hypothesis and conclude that the experimental results are reliable.

## E    DISTRIBUTION APPROXIMATION DETAILS

For each algorithm, we uniformly sample target strength value $v$ from the interval $[0, 1]$. Then we conduct each algorithm to obtain the trajectories and their corresponding actual attribute $u$. These values constitute the set $\mathcal{D}_{(u,v)}$. Let $u_{\max}$ and $u_{\min}$ denote the maximum and minimum values of $u$ in this set, respectively. Next, we divide the attribute strength interval and the actual attribute interval into equidistant segments, resulting in $K$ cells: $v \in [0, 1] = \bigcup_{i \in |K|} c_i^v$ and $u \in [u_{\min}, u_{\max}] = \bigcup_{i \in |K|} c_i^u$. We can then use the following equation to obtain the approximation:

$$\hat{P}_{ij}(u, v) = \frac{|\{(u, v)|u \in c_i^u, v \in c_j^v\}|}{|\mathcal{D}_{(u,v)}|} \approx \int_{c_i^u} \int_{c_j^v} p(u, v) \, du \, dv \tag{13}$$

## F    VIDEO CLIPS OF VARIOUS BEHAVIORS PRODUCED BY ALIGNDIFF

In this section, we aim to showcase additional behaviors generated by AlignDiff in the form of video segments. Specifically, we employ AlignDiff trained with human labels to generate behaviors. We focus on five attributes: movement speed, torso height, left-right leg preference, humanness defined in the Walker domain, and humanness defined in the Humanoid domain. For each attribute, we provide three video clips starting from a random initial state, with the corresponding attribute values adjusted to [0.9, 0.5, 0.1], respectively.

### F.1    WALKER: TORSO HEIGHT

As illustrated in Fig. 9, when the strength attribute value is 0.9, the walker exhibits minimal knee flexion to maintain a relatively higher torso position. At a value of 0.5, the walker moves in a normal manner. However, when the attribute value is 0.1, the walker adopts a posture close to kneeling, moving near the ground.

### F.2    WALKER: LEFT-RIGHT LEG PREFERENCE

As illustrated in Fig. 10, when the strength attribute value is 0.9, the walker primarily relies on the left leg for movement, while the right leg is scarcely utilized. At a value of 0.5, the walker alternates steps between the left and right legs. However, at 0.1, the walker predominantly relies on the right leg for movement, with minimal use of the left leg.

### F.3    WALKER: HUMANNESS

As illustrated in Fig. 11, when the strength attribute value is 0.9, the walker exhibits a running pattern similar to that of a human. At a value of 0.5, the walker displays a reliance on a single leg, resembling the movement of a person with an injured right leg. However, at 0.1, the walker engages in minimal leg movement, taking small and fragmented steps, deviating significantly from human-like walking.

### F.4    HUMANOID:HUMANNESS

As illustrated in Fig. 12, when the strength attribute value is 0.9, the humanoid is capable of walking with both legs, resembling human locomotion. At a strength value of 0.5, the humanoid can only perform single-leg movements, resembling a person with an injured right leg. When the strength value is 0.1, the humanoid will collapse and exhibit convulsive movements.

## G    IMPLEMENTATION DETAILS

In this section, we provide more implementation details of AlignDiff.

Table 11: Hyperparameters of the attribute strength model.

| Hyperparameter | Value |
|---|---|
| Embedding dimension | 128 |
| Number of attention heads | 4 |
| Number of layers | 2 |
| Dropout | 0.1 |
| Number of ensembles | 3 |
| Batch size | 256 |
| Optimizer | Adam |
| Learning rate | $10^{-4}$ |
| Weight decay | $10^{-4}$ |
| Gradient steps | 3000 |

Table 12: Hyperparameters of the diffusion model.

| Hyperparameter | Value |
|---|---|
| Embedding dimension of DiT | 384 Hopper/Walker
512 Humanoid |
| Number of attention heads of DiT | 6 Hopper/Walker
8 Humanoid |
| Number of DiT blocks | 12 Hopper/Walker
14 Humanoid |
| Dropout | 0.1 |
| Planning horizon | 32 Walker
100 Hopper/Humanoid |
| Batch size | 32 |
| Optimizer | Adam |
| Learning rate | $2 \times 10^{-4}$ |
| Weight decay | $10^{-4}$ |
| Gradient steps | $5 \times 10^5$ Hopper/Walker
$10^6$ Humanoid |

## G.1 ATTRIBUTE STRENGTH MODEL

- The attribute strength model utilizes a Transformer Encoder architecture without employing causal masks. We introduce an additional learnable embedding, whose corresponding output is then mapped to relative attribute strengths through a linear layer. The structural parameters of the Transformer and the training hyperparameters are presented in Table 11.

- We train a total of three ensembles, and during inference, the average output of the ensembles is taken as the final output.

- The direct output of the attribute strength model is a real number without range constraints, but the attribute strengths we define are within the range of 0 to 1. Therefore, we extract the maximum and minimum values of each attribute strength from the dataset of each task and normalize the output attribute values to be within the range of 0 to 1.

## G.2 DIFFUSION MODEL

- The AlignDiff architecture consists of a DiT structure comprising 12 DiT Blocks. The specific structural and training parameters can be found in Table 12.

- For the Walker benchmark, we use a planning horizon of 32, while for Hopper and Humanoid, the planning horizons are set to 100. We find that the length of the planning horizon should be

chosen to adequately capture the behavioral attributes of the agent, and this principle is followed in selecting the planning horizons for the three tasks in our experiments.

- We use 200 diffusion steps, but for Hopper and Walker, we only sample a subsequence of 10 steps using DDIM, and for Humanoid, we use a subsequence of 20 steps.

- We employ a guide scale of 1.5 for all tasks. We observed that smaller guide scales result in slower attribute switching but more reliable trajectories, while larger guide scales facilitate faster attribute switching but may lead to unrealizable trajectories.

---

**Algorithm 1** AlignDiff training

---

**Require:** Annotated Dataset $\mathcal{D}_G$, epsilon estimator $\epsilon_\phi$, unmask probability $p$
  **while** not done **do**
    $(\boldsymbol{x}_0, \boldsymbol{v^\alpha}) \sim \mathcal{D}_G$
    $t \sim \text{Uniform}(\{1, \cdots, T\})$
    $\epsilon \sim \mathcal{N}(\boldsymbol{0}, \boldsymbol{I})$
    $\boldsymbol{m^\alpha} \sim \mathcal{B}(k, p)$
    Update $\epsilon_\phi$ to minimize Eq. (4)
  **end while**

---

**Algorithm 2** AlignDiff planning

---

**Require:** epsilon estimator $\epsilon_\phi$, attribute strength model $\hat{\zeta}_\theta^\alpha$, target attribute strength $\boldsymbol{v^\alpha}$, attribute mask $\boldsymbol{m^\alpha}$, $S$ length sampling sequence $\kappa$, guidance scale $w$
  **while** not done **do**
    Observe state $s_t$; Sample $N$ noises from prior distribution $x_{\kappa_S} \sim \mathcal{N}(\boldsymbol{0}, \boldsymbol{I})$
    **for** $i = S, \cdots, 1$ **do**
      Fix $s_t$ for $x_{\kappa_i}$
      $\tilde{\epsilon}_\phi \leftarrow (1+w)\epsilon_\phi(x_{\kappa_i}, \kappa_i, \boldsymbol{v^\alpha}, \boldsymbol{m^\alpha}) - w\epsilon_\phi(x_{\kappa_i}, \kappa_i)$
      $x_{\kappa_{i-1}} \leftarrow \text{Denoise}(x_{\kappa_i}, \tilde{\epsilon}_\phi)$ // Eq. (5)
    **end for**
    $\tau \leftarrow \underset{\boldsymbol{x}_0}{\arg\min} \, ||(\boldsymbol{v^\alpha} - \hat{\zeta}_\theta^\alpha(\boldsymbol{x}_0)) \circ \boldsymbol{m^\alpha}||_2^2$
    Extract $a_t$ from $\tau$
    Execute $a_t$
  **end while**

---

## H EXTENSIVE ABLATION EXPERIMENTS

### H.1 ACCURACY OF THE ATTRIBUTE STRENGTH MODEL TRAINED ON HUMAN LABELS

Table 13: Success rate of the attribute strength model trained on human feedback datasets.

| Size of training sets | Speed | Torso height | Stride length | Left-right leg preference | Humanness |
|---|---|---|---|---|---|
| 3,200 | 92.48 | 92.68 | 83.01 | 82.42 | 87.60 |
| 1,600 | 91.09 | 90.54 | 83.22 | 81.32 | 84.22 |
| 800 | 91.80 | 91.41 | 82.81 | 82.62 | 79.30 |

The accuracy of the attribute strength model trained on human labels can help us understand the ability of the model for attribute alignment. Therefore, we conducted an additional experiment. We randomly selected 800 out of 4,000 human feedback samples from the Walker-H task as a test set. From the remaining 3,200 samples, we collected 3,200/1,600/800 samples as training sets, respectively, to train the attribute strength model. Subsequently, we recorded the highest prediction success rate of the model in the test set during the training process; see Table 13. We observed that only the "Humanness" attribute exhibited a significant decrease in prediction accuracy as the number of training samples decreased. This could be attributed to the fact that "Humanness" is the most abstract attribute among the predefined ones, making it more challenging for the model to learn discriminative patterns with limited feedback labels.

Table 14: Percentage of human labels that agree with the ground truth (annotated as 'gt') and with other annotators (annotated as 'inter'). "Masked agreement" indicates the agreement calculation after excluding samples labeled as "equivalent performance".

| Type | Speed | Torso height | Stride length | Left-right leg preference | Humanness |
|---|---|---|---|---|---|
| Agreement (gt) | 85.55 | 84.25 | – | – | – |
| Masked agreement (gt) | 96.96 | 98.16 | – | – | – |
| Agreement (inter) | 84 | 77 | 81 | 79 | 72 |
| Masked agreement (inter) | 99 | 91 | 93 | 97 | 86 |

Table 15: Relationship between inference time and performance.

| Sample steps | Inference time per action (seconds) | Performance |
|---|---|---|
| 10 | $0.286 \pm 0.003$ | $0.628 \pm 0.026$ |
| 5 | $0.145 \pm 0.002$ | $0.621 \pm 0.023$ |
| 3 | $0.089 \pm 0.002$ | $0.587 \pm 0.019$ |

## H.2 AGREEMENT OF THE ANNOTATORS

As annotator alignment is always a challenge in the RLHF community, we aimed to analyze the level of agreement achieved by the human labels collected through crowd-sourcing. Since the assigned samples for each annotator were completely different during our data collection, it was not possible to calculate interannotator agreement for this portion of the data. To address this, we selected two attributes, velocity and height, from the Walker task, for which ground truth values could be obtained from the MuJoCo engine. We calculated the agreement between the annotators and the ground truth for these attributes. And, to further assess inter-annotator agreement, we reached out to some of the annotators involved in the data collection process. Three annotators agreed to participate again, and we randomly selected 100 pairs of videos for them to annotate. We calculate the agreement among the three annotators (considering an agreement when all three annotators provided the same feedback, excluding cases where they selected "equivalent performance"). The results are presented in Table 14. We observed a high level of agreement among the annotators for attributes other than 'Humanness', which may be attributed to the relatively straightforward nature of these attributes and the provision of detailed analytical guidelines, as demonstrated in Appendix B. The agreement on the "Humanness" attribute was relatively lower, which can be attributed to its abstract nature and the varying judgment criteria among individuals. This observation is also supported by the phenomenon highlighted in Appendix H.1, where the attribute strength model showed increased sensitivity to the number of training labels for the "Humanness" attribute.

## H.3 THE EFFECT OF DIFFUSION SAMPLE STEPS

AlignDiff, like other diffusion planning methods such as Diffuser (Janner et al., 2022) and Decision Diffuser (DD) (Ajay et al., 2023), requires multiple steps of denoising to generate plans, which can lead to long inference times. However, AlignDiff has made efforts to mitigate this issue by taking advantage of the DDIM sampler (Song et al., 2021), which allows plan generation with fewer sampling steps. To investigate the relationship between inference time and AlignDiff performance, we conducted an additional experiment. Specifically, we evaluated different numbers of sampling steps on the Walker-S task, measuring the MAE area metric and the average time per step for decision-making, see Table 15. We observed that reducing the sampling steps from 10 (as used in our experiments) to 5 did not lead to a significant decline in performance, but halved the inference time. Notably, a noticeable performance decline occurred only when reducing the sampling steps to 3. Compared to Diffuser's use of 20 sampling steps and DD's use of 100 sampling steps, we have already minimized the inference time to the best extent possible within the diffusion planning approach. In future work, we will consider inference time as a crucial optimization objective and strive to further reduce it.

## H.4 THE EFFECT OF THE ATTRIBUTE-ORIENTED ENCODER

Table 16: Ablation of the usage of an attribute-oriented encoder.

| Label | Environment | AlignDiff (no enc.) | AlignDiff |
|---|---|---|---|
| Synthetic | Walker | $0.544 \pm 0.081$ | $0.621 \pm 0.023$ |

As mentioned in Section 4.3, setting the attribute strength value to 0 carries practical significance, as it represents the weakest manifestation of an attribute. Therefore, directly masking an attribute to indicate that it is not required may lead to confusion between "weakest manifestation" and "not required" by the network. In other words, providing an expected attribute value close to 0 might make it difficult for the network to differentiate whether it should exhibit a weak manifestation of the attribute or not require it. However, these scenarios are hypothetical, and we aim to supplement this observation with an ablation experiment to demonstrate this phenomenon quantitatively. We replace the attribute encoder of AlignDiff with a simple MLP and apply a mask (i.e., using 0 to indicate the absence of a required attribute) to the "not needed" attributes (denoted as AlignDiff (no enc.)). As shown in Table 16, the results on Walker-S reveal a noticeable performance drop. This suggests that the current masking approach in AlignDiff is effective.

## H.5 THE EFFECT OF THE NUMBER OF SELECTABLE TOKENS

Table 17: Area metric table of AlignDiff using different numbers of selectable tokens.

| Label | Environment | AlignDiff ($V$=10) | AlignDiff ($V$=50) | AlignDiff ($V$=100) |
|-------|-------------|-------------------|-------------------|---------------------|
| Synthetic | Hopper | $0.597 \pm 0.009$ | $0.626 \pm 0.025$ | $0.628 \pm 0.026$ |

In the AlignDiff attribute-oriented encoder, we discretize the strength values of all attributes, with each attribute assigned to one of the $V$ selectable tokens. As expected, reducing the value of $V$ can lead to an increase in the quantization error and a decrease in the precision of the preference alignment. To investigate the relationship between the performance of AlignDiff and the number of selectable tokens $V$, we conducted an additional experiment on the Hopper-S task. We tested the performance of AlignDiff with three different values of $V$ (10/50/100), and the results are shown in Table 17. We observed that when $V$ was reduced from 100 to 50, AlignDiff did not show a significant performance drop. However, a significant performance decline was observed when $V$ was reduced to 10. From Fig. 4, we can observe that only around 10% of the samples in the Hopper-S task achieved an MAE below 0.02 (the quantization error with $V$=50), while approximately 90% of the samples achieved an MAE below 0.1 (the quantization error with $V$=10). This suggests that we can strike a balance between parameter size and performance by analyzing the MAE curve to select an appropriate value for $V$.

## I HOW LANGUAGE MODEL USED IN THE PIPELINE

In the AlignDiff pipeline, the Sequence-Bert (Reimers & Gurevych, 2019) is used as an intermediary for transforming "natural language" to "attributes". Specifically, we keep an instruction corpus, where each element can be represented as a triplet $\{(\text{emb}, \text{attr}, \text{dir})\}$, where emb represents the Sentence-Bert embedding of a given language instruction (e.g., "Please run faster"), attr represents the attribute that the language instruction intends to modify (e.g., "speed"), and dir represents the direction of attribute change, either "increase" or "decrease."

When a human user provides a language instruction, it is first transformed into an embedding by Sentence-Bert. Then, cosine similarity is calculated between the embedding and all emb elements in the instruction set. The instruction with the highest similarity is considered as the user's intention, allowing us to determine whether the user wants to increase or decrease a specific attribute. For example, if the user wants to increase the "speed" attribute, we set $v^{\text{speed}}$ to $(v^{\text{speed}} + 1)/2$, and if wants to decrease it, we set $v^{\text{speed}}$ to $v^{\text{speed}}/2$. The current approach may be relatively simple, but it is still sufficient to capture some human intentions. For instance, Figure 6 on page 9 of the paper demonstrates full control solely based on natural language instructions. In the left image, we provided instructions ['Please move faster.', 'Keep increasing your speed.', 'You can slow down your pace now.'] at 200/400/600 steps, respectively. In future work, we would try to introduce LLM (Language Learning Model) to assist in recognizing more complex language instructions.

Table 18: Area metric table of AlignDiff and other two RBA-based methods.

| Label | Environment | RBA | RBA+ | AlignDiff |
|-------|-------------|-----|------|-----------|
| Synthetic | Walker | $0.293 \pm 0.026$ | $0.445 \pm 0.012$ | $\mathbf{0.621 \pm 0.023}$ |

## J COMPARISON WITH RBA

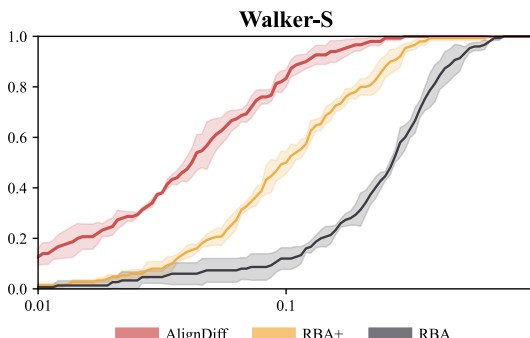

Figure 13: MAE curve of AlignDiff and other two RBA-based methods.

AlignDiff and RBA are two fundamentally different approaches. AlignDiff is a decision model that aligns behaviors via planning, whereas RBA mainly trains a reward model to represent human preferences, without direct decision-making capabilities. Even comparing the reward models, AlignDiff's attribute strength model has unique properties, such as evaluating variable-length trajectory attributes and supporting masks for uninterested data, which RBA's per-step reward function lacks. To fully illustrate the differences, we briefly revisit the RBA methodology (focusing on RBA-Global, which is most similar to AlignDiff's RLHF part).

RBA first trains an attribute-conditioned reward model using the same method as described in Appendix C.3, Eqs. (9) to (11), to support downstream policy learning for alignment. The original RBA paper uses a "decision-making" approach that does not require separate training, as noted in their Github repository[1]: *As mentioned in the paper, the current implementation optimizes the reward simply by sampling a large set of rollouts with the scripts or policies that we used to synthesize behavior dataset.* Specifically, the authors construct a policy library $\{\pi_i\}$ from all policies used to synthesize the behavior dataset. For a given state $s$, they search this library to find the action $a = \arg\max_{a_i \in \{\pi_i(s)\}} r_{\mathrm{RBA}}(s, a_i)$ that maximizes the RBA reward model. This approach is impractical, since constructing a sufficiently comprehensive policy library is difficult, and action optimization via search can be extremely time-consuming.

To construct a proper RBA baseline for comparison, we desire an approach that can align with human preferences whenever they change without separate training, like AlignDiff. So, as mentioned in Section 5.1, we use an offline attribute-conditioned TD learning algorithm (a modified TD3BC) to maximize the RBA reward model. Unlike RBA's search over a limited library, this approach is more practical (no need for a large policy library) and can generalize across diverse policies. Therefore, instead of lacking an RBA comparison, we actually compare against a refined version of RBA.

In this section, to quantitatively illustrate the difference, we further reconstruct the exact RBA decision process using the SAC policies, which are used to collect the Walker-S dataset, as the policy library, comparing it with AlignDiff and TDL (referred to as RBA+ here to emphasize it as an improved RBA). The resulting MAE curve and area table are presented in Fig. 13 and Table 18. The experiments confirm our expectations: AlignDiff outperforms RBA+, and RBA+ outperforms RBA due to its greater robustness. We offer an observation on RBA's poor performance: our specialized SAC policies behave erratically outside their specialized motion modes, a general RL issue. Since we use random initial states to test stability when behaviors must change, this is very challenging for a policy-library-search-based method RBA.

---

[1]https://github.com/GuanSuns/Relative-Behavioral-Attributes-ICLR-23/blob/main/README.md#step-4-interacting-with-end-users

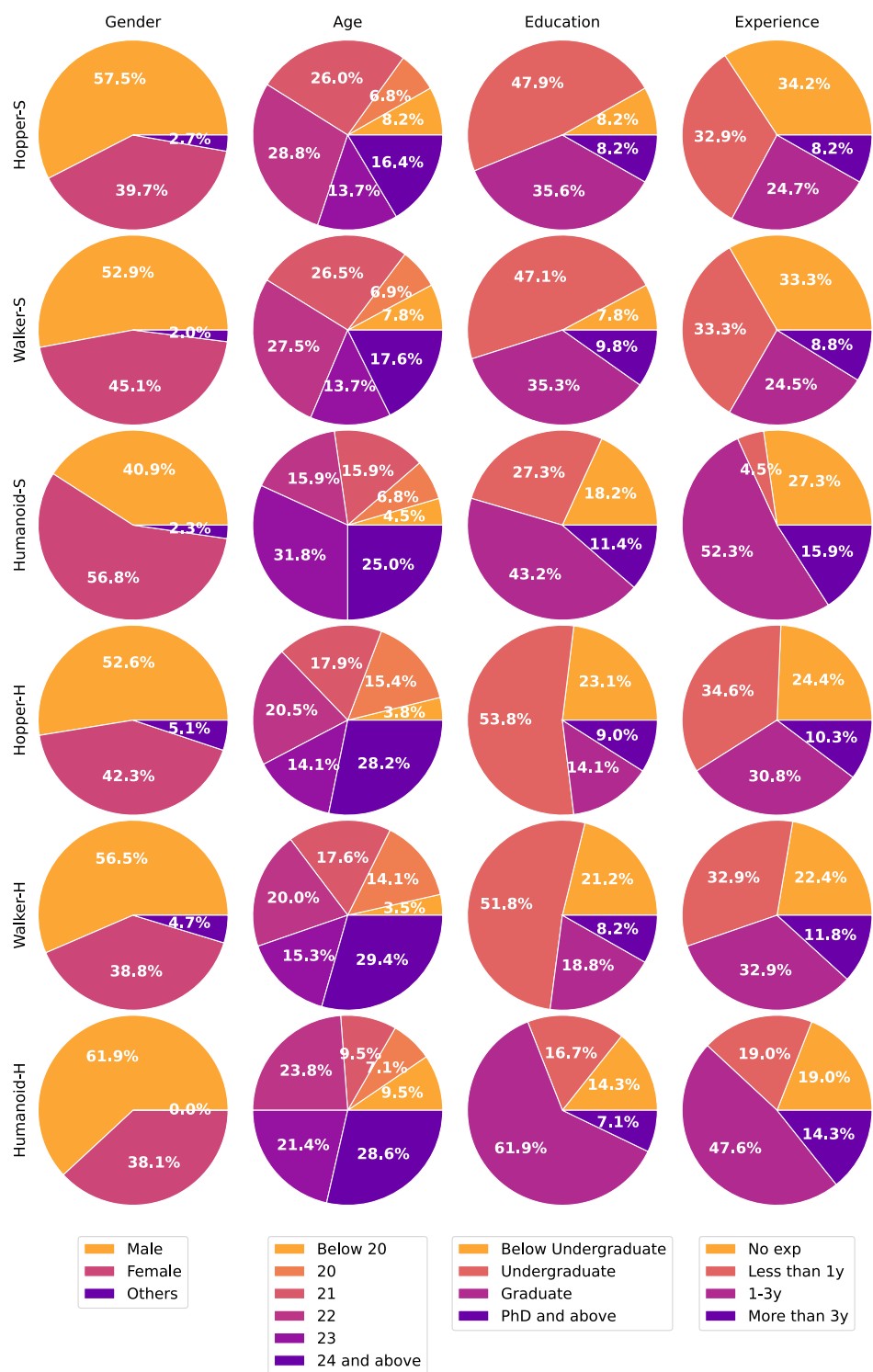

Figure 8: Distribution of participants involved in the human assessment experiment is presented. Prior to receiving the questionnaire, each participant is requested to provide basic information including gender, age, education, and AI experience. We present the distribution of these information categories in the form of pie charts.

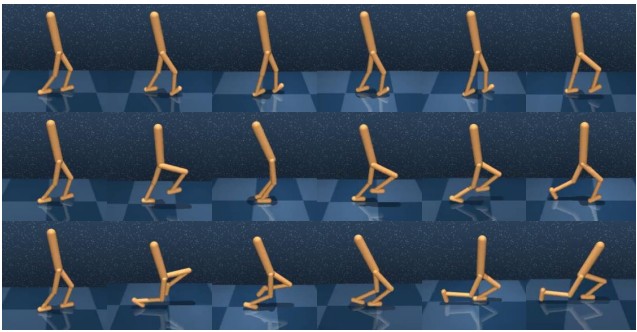

Figure 9: The behaviors generated by AlignDiff when only adjusting the torso height attribute. From the first row to the last row, they respectively represent attribute strength values of [0.9, 0.5, 0.1].

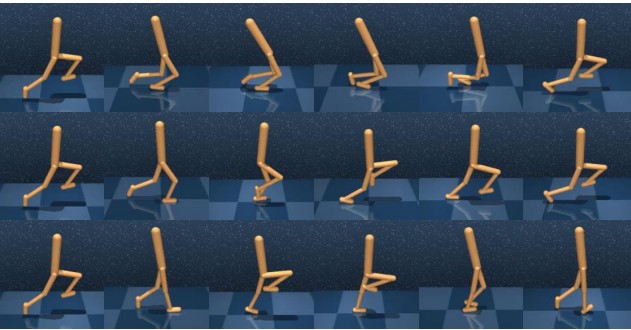

Figure 10: The behaviors generated by AlignDiff when only adjusting the left-right leg preference attribute. From the first row to the last row, they respectively represent attribute strength values of [0.9, 0.5, 0.1].

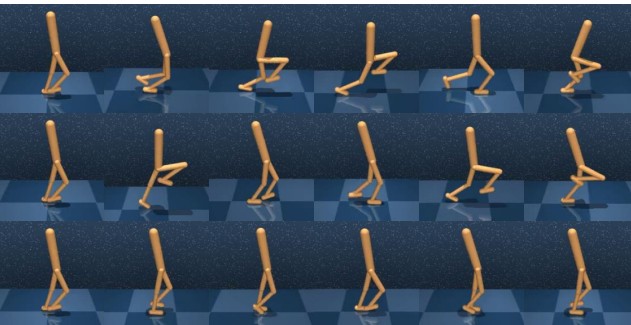

Figure 11: The behaviors generated by AlignDiff when only adjusting the humanness attribute. From the first row to the last row, they respectively represent attribute strength values of [0.9, 0.5, 0.1].

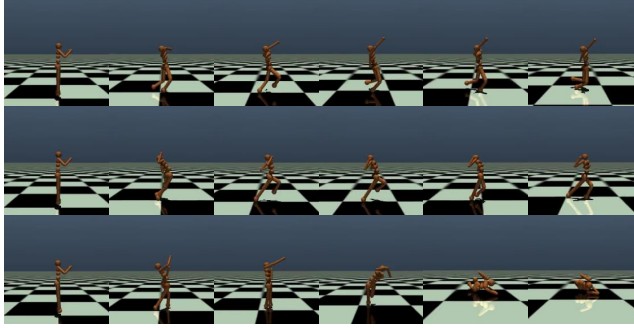

Figure 12: The behaviors generated by AlignDiff when only adjusting the humanness attribute. From the first row to the last row, they respectively represent attribute strength values of [0.9, 0.5, 0.1].

