# OpenReview forum: "AlignDiff: Aligning Diverse Human Preferences via Behavior-Customisable Diffusion Model"
_ICLR.cc/2024/Conference — ICLR 2024 poster_

### Official Review · Reviewer_pwJk · 2023-10-31

**Soundness:** 3 good
**Presentation:** 3 good
**Contribution:** 3 good
**Rating:** 8
**Confidence:** 2

**Summary:**

This paper proposes a diffusion model with RLHF to train an RL agent that follows human preferences and instructions. A attribute strength model is trained on a newly built human feedback datasets, which is leveraged to annotate the behavior dataset. Extensive experiments are conducted on the proposed method in terms of the preference matching, switching, and covering. All achieves superior performance compared to baselines.

**Strengths:**

1. The idea of using RLHF to align human preference is reasonable and insightful.
2. The experiments are extensive and verify the effectiveness of the proposed method.
3. The design of the attribute strength and the corresponding datasets could be helpful to many relative future works.

**Weaknesses:**

Could the authors offer more clarifications and analysis to demonstrate the extent to which the proposed attribute strength can encompass a broad spectrum of human preferences and instructions?
How accurate is the language model to find the correct attribute strength that match user's intent?

**Questions:**

Please refer to the weaknesses

---

> ### Author Response · Authors · 2023-11-19
> **Response to Reviewer pwJk**
>
> We sincerely appreciate the reviewer's recognition of our work, please see the following for our response.
>
> **(Q1) Could the authors offer more clarifications and analysis to demonstrate the extent to which the proposed attribute strength can encompass a broad spectrum of human preferences and instructions? How accurate is the language model to find the correct attribute strength that match user's intent?**
>
> We thank the reviewer for this highly inspiring question.
>
> (1) As mentioned in the paper, human preferences are complex and influenced by various intrinsic and extrinsic factors. We acknowledge that the predefined attributes presented in the paper may not necessarily meet the needs of all individuals. However, AlignDiff does not impose any limitations on the number of predefined attributes. In practice, designers can define a much larger set of attributes based on users' specific requirements or deployment scenarios. As the number of attributes increases, **the range of behaviors that the agent can exhibit expands exponentially**. Therefore, we believe that AlignDiff has the potential to accommodate the preferences of a significant number of users.
>
> (2) The current approach for the language model to find the correct attribute strength may be relatively simple, but it is still sufficient to capture some human intentions. For instance, **Figure 6 on page 9** of the paper demonstrates full control solely based on natural language instructions. In the left image, we provided instructions ['Please move faster.', 'Keep increasing your speed.', 'You can slow down your pace now.'] at 200/400/600 steps, respectively. In future work, we would explore the integration of LLMs. Leveraging their remarkable capabilities to assist the intent analysis.

---

### Official Review · Reviewer_WtmM · 2023-10-31

**Soundness:** 3 good
**Presentation:** 4 excellent
**Contribution:** 3 good
**Rating:** 8
**Confidence:** 3

**Summary:**

This paper focuses on the issue of consistency between agent behavior and human preferences in RLHF, and proposes an alignment method based on the diffusion model. The authors construct a multi-perspective human feedback dataset and train an attribute model, which is then used to relabel the dataset. A diffusion model is utilized as a planner and it's trained on the preference-aligned dataset. In this way, the authors achieve preference aligning between different human. Both quantitative and qualitative experimental results demonstrate the effectiveness of this method.

**Strengths:**

1. The authors revisit the impact of inherent human annotator preferences on reinforcement learning training, which is illuminating.

2. The proposed method is innovative and achieves relatively good results, which is demonstrated by the experiments.

3. The visualizations and supplementary material provided in the website support the paper and make it easier for readers to understand.

4. The paper is clearly written and well organized.

**Weaknesses:**

1. The paper does not include ablation experiments on attribute model training, so the actual effect of attribute alignment is not easy to measure.

2. The explanation of some details of the method is not clear enough. For example, the meaning of equation (5) and "inpainting manner" needs further clarification.

**Questions:**

See Weaknesses.

---

> ### Author Response · Authors · 2023-11-19
> **Response to Reviewer WtmM**
>
> We sincerely appreciate the reviewer's recognition of our work, please see the following for our response.
>
> **(Q1) The paper does not include ablation experiments on attribute model training, so the actual effect of attribute alignment is not easy to measure.**
>
> The attribute model is a crucial component in capturing human preferences and agent behavior. To enable the model to learn preferences effectively, all algorithms in our experiments utilized the same attribute model. Consequently, we were unable to perform an ablation study on this particular component. However, we acknowledge the reviewer's concern regarding attribute alignment. We speculate that the reviewer is interested in understanding whether the model truly comprehends human preferences, and the predictive accuracy of the attribute model should also reflect this aspect of the problem. Therefore, we conducted the following additional experiments:
>
> We randomly selected 800 out of 4,000 human feedback samples in Walker as a test set. From the remaining 3,200 samples, we collected 3,200/1,600/800 samples as training sets, respectively, to train the reward model. We then recorded the highest prediction accuracy of the model on the test set during the training process. The results are as follows:
>
> | Size of training sets | Speed | Torso height | Stride length | Left-right leg preference | Humanness |
> | --------------------- | ----- | ------------ | ------------- | ------------------------- | --------- |
> | 3,200                 | 92.48 | 92.68        | 83.01         | 82.42                     | 87.60     |
> | 1,600                 | 91.09 | 90.54        | 83.22         | 81.32                     | 84.22     |
> | 800                   | 91.80 | 91.41        | 82.81         | 82.62                     | 79.30     |
>
> We found that our attribute model can indeed predict human preferences with a high level of accuracy.. For this part, we add an extension discussion in the revision (**see Appendix H.1, page 22, revised version paper**)
>
>
>
> **(Q2) The explanation of some details of the method is not clear enough. For example, the meaning of equation (5) and "inpainting manner" needs further clarification.**
>
> We apologize for any confusion caused by our unclear description. Due to the limitations of the main text, some details may have been omitted for brevity. **Equation (5) on page 6** of the paper represents the reverse process of DDIM, and the symbols used in this equation are explained in detail in **Section 3, page 3, DDIM part**. Specifically, starting from noisy data, we can continuously denoise the data using the trained noise predictor through Equation (5) to obtain candidate trajectories. The term "inpainting manner" refers to a commonly used approach in diffusion planning. During the denoising process, since the current state $s_t$ is known, we fix $s_t$ similar to inpainting in image generation tasks, where a portion of the image is known and the remaining part needs to be completed. Now we have revised the description to "During the process, we fix the current state $s_t$ as it is always known" to make it clearer. The specific form of this operation is described in **Algorithm 2, page 22** of the paper.

---

### Official Review · Reviewer_t4Ww · 2023-11-01

**Soundness:** 3 good
**Presentation:** 3 good
**Contribution:** 2 fair
**Rating:** 6
**Confidence:** 3

**Summary:**

This paper presents a method for aligning agent behaviors with human preferences. Firstly, it introduces multi-perspective human feedback datasets. Secondly, it trains an attribute-conditioned diffusion model, referred to as AlignDiff, to act as a director for preference alignment during the inference phase. AlignDiff utilizes Reinforcement Learning from Human Feedback (RLHF) to quantify human preferences, allowing it to match user behaviors and seamlessly transition between different preferences.

**Strengths:**

1. The proposed diffusion-based framework demonstrates exceptional performance in decision-making scenarios involving complex dynamics.
2. The method presented in this paper is capable of effectively matching user-customized behaviors and seamlessly transitioning between different preferences.
3. Additionally, this paper introduces a valuable contribution in the form of a multi-perspective human feedback dataset. This dataset has the potential to facilitate the wider adoption of human preference aligning techniques.
4. The proposed method leverages a multi-stage diffusion process, which effectively simulates the reinforcement learning (RL) process.

**Weaknesses:**

1. Both the proposed method and RBA utilize RLHF for aligning agent behaviors with human preferences. The novelty is unclear.
2. The diffusion model usually achieves the best result in the final step. How does the diffusion model guarantee the best human preference at each step? Does the proposed method obtain a plan with T diffusion steps? If so, how about the inference time?
3. The proposed method only did some ablation studies and has not compared with the state-of-the-art methods, such as RBA.

**Questions:**

1. What is the inference time?
2. How about the comparison with the state-of-the-art methods, such as RBA.

---

> ### Author Response · Authors · 2023-11-19
> **Response to Reviewer t4Ww (Part 1/2)**
>
> We thank the reviewer for the insightful and useful feedback, please see the following for our response.
>
> **(Q1) Both the proposed method and RBA utilize RLHF for aligning agent behaviors with human preferences. The novelty is unclear.**
>
> We would like to clarify that AlignDiff and RBA are **fundamentally different** approaches.
>
> 1. AlignDiff is a **decision framework that aligns behaviors via planning**, whereas RBA mainly trains a **reward model** only for representing human preferences, without direct decision-making capabilities.
>
> 2. AlignDiff's attribute strength model has unique properties, such as evaluating variable-length trajectory attributes and supporting masks for uninterested data, which RBA's per-step reward function lacks.
>
> 3. If users intend to deploy RBA for decision-making, they either need to (1) prepare a large strategy library in advance for RBA to search from, or (2) learn a new RL strategy from scratch for each new requirement. However, this approach fails to accommodate the mutability of human preferences, as mentioned in the Introduction section. On the contrary, AlignDiff can adapt to human preference in a zero-shot manner.
>
> 4. We show extra experiment results and explain in detail the different experimental approaches used by RBA and Aligndiff in reply to the Q2.
>
> **(Q2) The proposed method only did some ablation studies and has not compared with the state-of-the-art methods, such as RBA.**
>
> We apologize for any confusion caused and we add an extension discussion in the revision (**see Appendix J, page 25, revised version paper**).
>
> 1. **We indeed compared AlignDiff with an improved version of RBA, called TDL, as our baseline.** As mentioned in the response to Q1, RBA is impractical for decision-making purposes. To construct an improved version of RBA as a baseline for comparison, we use an offline attribute-conditioned TD learning algorithm (a modified TD3BC) to maximize the RBA reward model. Unlike RBA's search over a limited library, this approach is more practical (no need for a large policy library) and can generalize across diverse policies. Therefore, instead of lacking an RBA comparison, we actually compare against a refined version of RBA.
>
> 2. **Why not compare with the vanilla RBA?** To answer this question, we need to recall how RBA works. RBA first trains an attribute-conditioned reward model using the same method as described in **Appendix C.3, Equation (9-11)**, to support downstream policy learning for alignment. The original RBA paper uses a "decision-making" approach that does not require separate training, as noted in their public code [1]: ***As mentioned in the paper, the current implementation optimizes the reward simply by sampling a large set of rollouts with the scripts or policies that we used to synthesize behavior dataset**.* Specifically, the authors construct a policy library $\{\pi_i\}$ from all policies used to synthesize the behavior dataset. For a given state $s$, they search this library to find the action $a=\arg\max_{a_i\in\{\pi_i(s)\}}r_{\text{RBA}}(s,a_i)$ that maximizes the RBA reward model. This approach is impractical, since constructing a sufficiently comprehensive policy library is not useful, and action optimization via search can be extremely time-consuming.
>
> 3. **What if we build large policy library and compare AlignDiff with vanilla RBA?** To better answer this question and quantitatively illustrate the difference, we further reconstruct the exact **vanilla RBA** decision process using the SAC policies, which are used to collect the Walker-S dataset, as the policy library, comparing it with **AlignDiff** and TDL (referred to as **RBA+** here to emphasize it as an improved RBA). The resulting MAE curve and area table are presented in **Appendix J, page 25, revised version paper**:
>
> | Environment | RBA             | RBA+            | AlignDiff              |
> | ----------- | --------------- | --------------- | ---------------------- |
> | Walker      | $0.293\pm0.026$ | $0.445\pm0.012$ | $0.621\pm0.023$ |
>
> The experiments confirm our expectations: **AlignDiff outperforms RBA+, and RBA+ outperforms RBA** due to its greater robustness. We offer an observation on RBA's poor performance: our specialized SAC policies behave erratically outside their specialized motion modes, a general RL issue. Since we use random initial states to test stability when behaviors must change, this is very challenging for a policy-library-search-based method RBA.

---

> > ### Comment · Reviewer_t4Ww · 2023-11-22
> >
> > The rebuttal has addressed my concerns. Therefore, I improved my score to marginally above the threshold.

---

> ### Author Response · Authors · 2023-11-19
> **Response to Reviewer t4Ww (Part 2/2)**
>
> **(Q3) The diffusion model usually achieves the best result in the final step. How does the diffusion model guarantee the best human preference at each step? Does the proposed method obtain a plan with T diffusion steps? If so, how about the inference time?**
>
> Thank you for the question. AlignDiff, like other diffusion planning methods such as Diffuser [2] and Decision Diffuser (DD) [3], requires multiple steps of denoising to generate the best plans, which can lead to long inference times. However, AlignDiff has made efforts to mitigate this issue by taking advantage of the DDIM sampler [4], which allows plan generation with fewer sampling steps. To investigate the relationship between inference time and AlignDiff performance, we conducted an additional experiment. Specifically, we evaluated different numbers of sampling steps on the Walker-S task, measuring the MAE area metric and the average time per step for decision-making, see the table below:
>
> | Sample steps  | Inference time per step (seconds) | Performance   |
> | ------------- | --------------------------------- | ------------- |
> | 10            | $0.286 ± 0.003$                   | 0.628 ± 0.026 |
> | 5             | $0.145 ± 0.002$                   | 0.621 ± 0.023 |
> | 3             | $0.089 ± 0.002$                   | 0.587 ± 0.019 |
> | 20 (Diffuser) | $\approx 0.57$                    |               |
> | 100 (DD)      | $\approx 2.86$                    |               |
>
> We observed that reducing the sampling steps from 10 (as used in our experiments) to 5 did not lead to a significant decline in performance, but halved the inference time. Notably, a noticeable performance decline occurred only when reducing the sampling steps to 3. Compared to Diffuser's use of 20 sampling steps and DD's use of 100 sampling steps, we have already minimized the inference time to the best extent possible within the diffusion planning approach. In future work, we will consider inference time as a crucial optimization objective and strive to further reduce it. For this part, we add an extension discussion in the revision (**see Appendix H.3, page 23, revised version paper**)
>
>
>
> References:
>
> [1] The official implementation of RBA on GitHub. https://github.com/GuanSuns/Relative-Behavioral-Attributes-ICLR-23
>
> [2] Michael Janner, Yilun Du, Joshua Tenenbaum, and Sergey Levine. Planning with diffusion for flexible behavior synthesis. In International Conference on Machine Learning, ICML, 2022.
>
> [3] Anurag Ajay, Yilun Du, Abhi Gupta, Joshua B. Tenenbaum, Tommi S. Jaakkola, and Pulkit Agrawal. Is conditional generative modeling all you need for decision making? In The Eleventh International Conference on Learning Representations, ICLR, 2023.
>
> [4] Jiaming Song, Chenlin Meng, and Stefano Ermon. Denoising diffusion implicit models. In Interna-
> tional Conference on Learning Representations, ICLR, 2021.

---

### Official Review · Reviewer_SGJ2 · 2023-11-04

**Soundness:** 3 good
**Presentation:** 3 good
**Contribution:** 3 good
**Rating:** 6
**Confidence:** 3

**Summary:**

This paper utilizes the diffusion model conditioned on attributes for planning. Inspired by RLHF, this paper first fits a reward model trained to predict preference over trajectories given human preference, followed by using it to predict the attribute strengths given the episode. This is then used to label the unlabelled episodes with the corresponding attributes, which are then used to condition the diffusion model, trained in a classifier-free guidance sense.

The paper shows the advantages of the proposed technique over existing baselines for learning the policy including on human preferences, and tests its robustness when the strengths are suddenly changed in between. Lastly, an ablation of label pool size is done to understand its impact on learning the reward model.

**Strengths:**

The paper is easy to follow in most places, and I like the experiments on the robustness and label efficiency of the reward function.

**Weaknesses:**

As someone who is not well versed in the empirical reinforcement community, my weak comments would be high level and mostly some of my confusion throughout the paper.

- Can authors provide the inter-annotator correlation when they're labeling the reward model? Annotator alignment is a problem in the RLHF community when it comes to LLMs and even RLHF for diffusion models in image generation (See [1]) therefore it would be good to see some numbers on annotator agreement.

- Can authors provide the accuracy of the reward model after training it based on human preference? Does the Area metric in Table 5 correspond to that? Moreover, what is the accuracy of random guessing? I am assuming it would be 50%?

- Are there any ablations done in case one does not apply masking in the way current AlignDiff is applying? What if one just used 0s in the strength where it is not needed? I think having that result would further showcase the utility of masking in the current way.

- How exactly is BERT used in the pipeline? How is the mapping from BERT representations to strength and mask learned?

- How is the performance affected by the precision of discretization?

- $\mathcal{B}(k, p)$ is not defined.

- How is the performance of the reward function (that predicts strengths) affected by the length of episodes, when varied during training and inference time (zero-shot say)?


[1] Human Preference Score v2: A Solid Benchmark for Evaluating Human Preferences of Text-to-Image Synthesis.

**Questions:**

Refer to the weakness section.

---

> ### Author Response · Authors · 2023-11-19
> **Response to Reviewer SGJ2 (Part 1/3)**
>
> We thank the reviewer for the insightful and useful feedbacks, please see the following for our response.
>
> **(Q1) Can authors provide the inter-annotator correlation when they're labeling the reward model?**
>
> Yes, we can. Since the assigned samples for each annotator were completely different during our data collection, it was not possible to calculate inter-annotator agreement for this portion of the data. To address this, we selected two attributes, velocity and height, from the Walker task, for which ground truth values could be obtained from the MuJoCo engine. We calculated the agreement between the annotators and the ground truth for these attributes. And, to further assess inter-annotator agreement, we reached out to some of the annotators involved in the data collection process. Three annotators agreed to participate again, and we randomly selected 100 pairs of videos for them to annotate. We calculated the agreement among the three annotators (considering an agreement when all three annotators provided the same feedback, excluding cases where they selected “equivalent performance”). The results are presented in the table below, which includes the percentage of human labels that agree with the ground truth (annotated as 'gt') and with other annotators (annotated as 'inter'). "Masked agreement" indicates the agreement calculation after excluding samples labeled as "equivalent performance".
>
> | Type                     | Speed | Torso height | Stride length | Left-right leg preference | Humanness |
> | ------------------------ | ----- | ------------ | ------------- | ------------------------- | --------- |
> | Agreement (gt)           | 85.55 | 84.25        | -             | -                         | -         |
> | Masked agreement (gt)    | 96.96 | 98.16        | -             | -                         | -         |
> | Agreement (inter)        | 84    | 77           | 81            | 79                        | 72        |
> | Masked agreement (inter) | 99    | 91           | 93            | 97                        | 86        |
>
> For this part, we add an extension discussion in the revision (**see Appendix H.2, page 23, revised version paper**)
>
>
>
> **(Q2) Can authors provide the accuracy of the reward model after training it based on human preference? Does the Area metric in Table 5 correspond to that? Moreover, what is the accuracy of random guessing? I am assuming it would be 50%?**
>
> 1. Yes, we can. We conducted the following additional experiments to show the prediction accuracy of the reward model (also known as the attribute strength model in our paper) trained with different numbers of human labels.
>
>    We randomly selected 800 out of 4,000 human feedback samples in Walker as a test set. From the remaining 3,200 samples, we collected 3,200/1,600/800 samples as training sets, respectively, to train the reward model. We then recorded the highest prediction accuracy of the model on the test set during the training process. The results are as follows:
>
> | Size of training sets | Speed | Torso height | Stride length | Left-right leg preference | Humanness |
> | --------------------- | ----- | ------------ | ------------- | ------------------------- | --------- |
> | 3,200                 | 92.48 | 92.68        | 83.01         | 82.42                     | 87.60     |
> | 1,600                 | 91.09 | 90.54        | 83.22         | 81.32                     | 84.22     |
> | 800                   | 91.80 | 91.41        | 82.81         | 82.62                     | 79.30     |
> | random guessing       | 33.33 | 33.33        | 33.33         | 33.33                     | 33.33     |
>
> ​		We found that only the "Humanness" attribute significantly decreased prediction accuracy as the number of training samples decreased. This may be because "Humanness" is the most abstract attribute among the predefined ones, and it becomes more challenging for the model to learn the discriminative patterns with fewer feedback labels.
>
> 2. The area metric in Table 5 does not directly reflect the prediction accuracy of the reward model. It merely reflects the influence of the number of training labels on the model. However, the results in Table 5 do align with the findings of the supplementary ablation experiment here. Even with a small number of labels, the model can accurately predict human preferences with high accuracy. Reducing the number of labels does not significantly compromise the model's performance.
>
> 3. Due to the three possible feedback options from humans for any given attribute (trajectory 1 performs stronger, equal performance, trajectory 2 performs stronger), the accuracy of random guessing should be **33%**. This result has also been included in the table for reference.
>
> For this part, we add an extension discussion in the revision (**see Appendix H.1, page 22, revised version paper**)

---

> ### Author Response · Authors · 2023-11-19
> **Response to Reviewer SGJ2 (Part 2/3)**
>
> **(Q3) Are there any ablations done in case one does not apply masking in the way current AlignDiff is applying? What if one just used 0s in the strength where it is not needed? I think having that result would further showcase the utility of masking in the current way.**
>
> Thanks for your helpful suggestion. As mentioned in **Section 4.3, page 5**, setting the attribute strength value to 0 carries practical significance, as it represents the weakest manifestation of an attribute. Therefore, directly masking an attribute to indicate that it is not required may lead to confusion between "weakest manifestation" and "not required" by the network. In other words, providing an expected attribute value close to 0 might make it difficult for the network to differentiate whether it should exhibit a weak manifestation of the attribute or not require it. To further showcase the utility of masking in the current way,  we conducted an ablation experiment. We replace the attribute encoder of AlignDiff with a simple MLP and apply a mask (i.e., using 0 to indicate the absence of a required attribute) to the "not needed" attributes (denoted as AlignDiff (no enc.)). We show the performance comparison in the following table:
>
> | Environment | AlignDiff (no enc.) | AlignDiff       |
> | ----------- | ------------------- | --------------- |
> | Walker      | $0.544\pm0.081$     | $0.621\pm0.023$ |
>
>
> The results on Walker-S reveal a noticeable performance drop. This suggests that the current masking approach in AlignDiff is effective.  For this part, we add an extension discussion in the revision (**see Appendix H.4, page 23, revised version paper**)
>
>
>
> **(Q4) How exactly is BERT used in the pipeline? How is the mapping from BERT representations to strength and mask learned?**
>
> Thank you for bringing up this question. Due to the limitations of the main text, we indeed provided a relatively concise explanation of this issue, which may have caused your confusion. In the current pipeline, the Sequence-Bert is used as an intermediary for transforming "natural language" to "attributes". Specifically, we keep an instruction corpus, where each element can be represented as a triplet $\{(\text{emb},\text{attr},\text{dir})\}$, where $\text{emb}$ represents the Sentence-Bert embedding of a given language instruction (e.g., "Please run faster"), $\text{attr}$ represents the attribute that the language instruction intends to modify (e.g., "speed"), and $\text{dir}$ represents the direction of attribute change, either "increase" or "decrease."
>
> When a human user provides a language instruction, it is first transformed into an embedding by Sentence-Bert. Then, cosine similarity is calculated between the embedding and all $\text{emb}$ elements in the instruction set. The instruction with the highest similarity is considered as the user's intention, allowing us to determine whether the user wants to increase or decrease a specific attribute. For example, if the user wants to increase the "speed" attribute, we set $v^{\text{speed}}$ to $(v^{\text{speed}}+1)/2$, and if wants to decrease it, we set $v^{\text{speed}}$ to $v^{\text{speed}}/2$. The current approach may be relatively simple, but it is still sufficient to capture some human intentions. For instance, Figure 6 on page 9 of the paper demonstrates full control solely based on natural language instructions. In the left image, we provided instructions ['Please move faster.', 'Keep increasing your speed.', 'You can slow down your pace now.'] at 200/400/600 steps, respectively. In future work, we would try to introduce LLM (Language Learning Model) to assist in recognizing more complex language instructions.
>
> For this part, we add an extension discussion in the revision (**see Appendix I, page 24, revised version paper**)

---

> ### Author Response · Authors · 2023-11-19
> **Response to Reviewer SGJ2 (Part 3/3)**
>
> **(Q5) How is the performance affected by the precision of discretization?**
>
> As anticipated, reducing the value of $V$ can lead to increased quantization error and a decrease in preference aligning precision. As you suggested, to further investigate the relationship between the performance of AlignDiff and the number of selectable tokens $V$, we conducted an additional experiment on the Hopper-S task. We tested the performance of AlignDiff with three different values of $V$ (10/50/100), and the results are shown in the following table:
>
> | Environment | AlignDiff ($V$=10) | AlignDiff ($V$=50) | AlignDiff ($V$=100) |
> | ----------- | ------------------ | ------------------ | ------------------- |
> | Hopper      | $0.597\pm0.009$    | $0.626\pm0.025$    | $0.628\pm0.026$     |
>
> We observed that when $V$ was reduced from 100 to 50, AlignDiff did not show a significant performance drop. However, a significant performance decline was observed when $V$ was reduced to 10. From **Figure 4, Section 4.4, page 6,** we can observe that only around 10% of the samples in the Hopper-S task achieved an MAE below 0.02 (the quantization error with $V$=50), while approximately 90% of the samples achieved an MAE below 0.1 (the quantization error with $V$=10). This suggests that we can strike a balance between parameter size and performance by analyzing the MAE curve to select an appropriate value for $V$. For this part, we add an extension discussion in the revision (**see Appendix H.5, page 24, revised version paper**)
>
>
>
> **(Q6) $\mathcal B(k,p)$ is not defined.**
>
> We sincerely apologize for not explicitly describing the notation for this distribution in the main text. The notation $\mathcal B(k,p)$ represents a binomial distribution, where $p$ denotes the unmask probability, which is the probability of sampling 1 from this distribution. The parameter $k$ represents the number of attributes, which is the number of independent and identically distributed trials in the distribution. The definition has been supplemented in the revision (**see Section 4.3, page 5, above Equation (4), revised version paper**)
>
>
>
> **(Q7) How is the performance of the reward function (that predicts strengths) affected by the length of episodes, when varied during training and inference time (zero-shot say)?**
>
> It is a good question! This question aligns with the rationale behind the design of the reward model structure described in our paper. In our experiments, the trajectory lengths inputted to the reward model **during the training and inference stages differ**. During training, we use a trajectory length of **100** (approximately equivalent to a 30-frame video spanning 3.3 seconds) to facilitate human annotators' assessment. However, during the inference stage, specifically in Walker's Dataset annotating phase, the input length is set to **32**, matching the planning length. We have observed that the structure we designed **can adapt to different input lengths**, allowing for the flexibility to change the input length during the inference stage. We also noticed that shorter input lengths lead to faster changes in behavior but with greater instability. Conversely, longer input lengths result in slower changes in behavior but with greater stability. In practice, we found that the length should be chosen to adequately capture the behavioral attributes of the agent. A related description can be found in **Appendix G.2, page 21.**

---

### Author Response · Authors · 2023-11-20
**Summarization of the first revised version**

We would like to express our gratitude to each reviewer for their insightful and helpful feedback. We have provided corresponding responses to each reviewer. Now, we summarize the updates made in the revised version of the paper to facilitate reviewers in finding the new content of interest:

- **Section 4.3, page 5:** Above Equation (4), an explanation of the distribution $\mathcal B(k,p)$ has been added.
- **Section 4.4, page 6:** Below Equation (5), the description has been modified to eliminate the use of "inpainting manner" for better reader understanding.
- **Appendix H, page 22:** A new appendix section has been added, expanding on additional ablation experiments.
  - **Appendix H.1, page 22:** Added an experiment that compares the accuracy of human preference prediction using attribute strength models trained with different numbers of human labels.
  - **Appendix H.2, page 23:** Added an experiment that supplements the agreement among human annotators.
  - **Appendix H.3, page 23:** Added an experiment that compares the decision-making time consumption and performance of AlignDiff under different diffusion sample steps.
  - **Appendix H.4, page 23:** Added an experiment that ablates the attribute encoder used in AlignDiff and compares it with a regular MLP encoder.
  - **Appendix H.5, page 24:** Added an experiment that compares the performance of AlignDiff with different numbers of tokens used for attribute discretization.
- **Appendix I, page 24:** Added a new appendix section that provides a detailed explanation of the usage of the language model in AlignDiff and further discussion.
- **Appendix J, page 25:** Added a new appendix section that provides a detailed explanation of the relationship between AlignDiff and RBA, clarifying that AlignDiff is compared with an improved version of RBA as a baseline in the experiments. This section also includes a performance comparison between AlignDiff and vanilla RBA.

These are all the additions in the revised version. We sincerely hope that the reviewers will continue to raise questions, and we will try to address them to the best of our ability.

---

### Meta-Review · Area_Chair_9p3g · 2023-12-06

**Metareview:**

Summary: This paper proposes AlignDiff, a method for using RLHF to guide diffusion model planning. The approach first uses a dataset of human preferences over diverse trajectories to predict attributes (e.g., velocity, step size, humanness) of an unlabelled trajectory; attributes are then used to guide the conditional diffusion model.

Strengths: The paper is clearly written, studies an interesting and relevant problem, and–particularly after the rebuttal–does a thorough job of evaluating and ablating the proposed method, showing favorable results.

Weaknesses: Most of the weaknesses raised by reviewers were addressed but one limitation of the method is the need to design a set of attributes, raising the question of how do you know these attributes cover the space of all things that matter to the end-users?

**Justification For Why Not Higher Score:**

While the paper is a well-written and presents a clear idea with value, I recommend a poster because the results are only demonstrated on fairly standard toy simulation environments like MuJoCo and require hand-engineering attributes for these specific environments.

**Justification For Why Not Lower Score:**

I agree with the reviewers that the overall approach is sound, the paper is clearly written, and well evaluated.

---

### Decision · Program_Chairs · 2024-01-16

Accept (poster)